# Analysis of the *Leishmania mexicana* promastigote cell cycle using imaging flow cytometry provides new insights into cell cycle flexibility and events of short duration

Jessie Howell[1], Sulochana Omwenga[2], Melanie Jimenez[3], Tansy C. Hammarton[2]*

**1** James Watt School of Engineering, University of Glasgow, Glasgow, United Kingdom, **2** School of Infection and Immunity, University of Glasgow, Glasgow, United Kingdom, **3** Biomedical Engineering Department, University of Strathclyde, Glasgow, United Kingdom

* Tansy.Hammarton@glasgow.ac.uk

**Data Availability Statement:** All relevant data are within the manuscript and its Supporting Information files.

## Abstract

Promastigote *Leishmania mexicana* have a complex cell division cycle characterised by the ordered replication of several single-copy organelles, a prolonged S phase and rapid G2 and cytokinesis phases, accompanied by cell cycle stage-associated morphological changes. Here we exploit these morphological changes to develop a high-throughput and semi-automated imaging flow cytometry (IFC) pipeline to analyse the cell cycle in live *L. mexicana*. Firstly, we demonstrate that, unlike several other DNA stains, Vybrant™ Dye-Cycle™ Orange (DCO) is non-toxic and enables quantitative DNA imaging in live promastigotes. Secondly, by tagging the orphan spindle kinesin, KINF, with mNeonGreen, we describe KINF's cell cycle-dependent expression and localisation. Then, by combining manual gating of DCO DNA intensity profiles with automated masking and morphological measurements of parasite images, visual determination of the number of flagella per cell, and automated masking and analysis of mNG:KINF fluorescence, we provide a newly detailed description of *L. mexicana* promastigote cell cycle events that, for the first time, includes the durations of individual G2, mitosis and post-mitosis phases, and identifies G1 cells within the first 12 minutes of the new cell cycle. Our custom-developed masking and gating scheme allowed us to identify elusive G2 cells and to demonstrate that the CDK-inhibitor, flavopiridol, arrests cells in G2 phase, rather than mitosis, providing proof-of-principle of the utility of IFC for drug mechanism-of-action studies. Further, the high-throughput nature of IFC allowed the close examination of promastigote cytokinesis, revealing considerable flexibility in both the timing of cytokinesis initiation and the direction of furrowing, in contrast to the related kinetoplastid parasite, *Trypanosoma brucei* and many other cell types. Our new pipeline offers many advantages over traditional methods of cell cycle analysis such as fluorescence microscopy and flow cytometry and paves the way for novel high-throughput analysis of *Leishmania* cell division.

**Funding:** This work was supported by the Engineering and Physical Sciences Research Council (PhD studentship no. EP/R513222/1 held by J.H.), the Royal Academy of Engineering (Research Fellowship no. RF/201718/1741 awarded to M.J.). and the Cunningham Trust (PhD studentship grant (PhD-CT-19-14) awarded to T.H. and M.J., and held by S.O.). There was no additional external funding received for this study. The funders had no role in study design, data collection and analysis, decision to publish, or preparation of the manuscript.

**Competing interests:** The authors have declared that no competing interests exist.

## Introduction

*Leishmania* spp. are unicellular parasitic protists responsible for the leishmaniases, a heterogenous group of neglected tropical diseases that affect humans and animals, causing disfiguring and potentially disabling cutaneous lesions to fatal visceral disease. Over 20 species of *Leishmania* can cause disease in humans, and there are more than 90 sand fly vector species that transmit the parasites between hosts [1]. Key to being able to complete its digenetic life cycle and to cause disease in humans and animals, is the ability of the parasite to proliferate in its hosts. Thus, understanding better how its cell cycle is regulated at the molecular level could offer new avenues for the development of novel therapeutics for leishmaniasis.

*Leishmania* are members of the Kinetoplastida, an order characterised by the presence of two DNA-containing organelles: the nucleus, and the kinetoplast, which contains the genome of the organism's single mitochondrion. The kinetoplast is linked to a pair of basal bodies comprising one immature pro-basal body and one mature basal body that subtends a single flagellum. The flagellum emerges from the cell body via a specialized invagination of the plasma membrane termed the flagellar pocket, which likely acts as the sole site of endo- and exocytosis in the cell [2]. *Leishmania* also possess a single Golgi apparatus, located between the nucleus and kinetoplast, and beneath the plasma membrane is a subpellicular microtubule cytoskeleton that remains assembled throughout the cell cycle. Replication and segregation of the single copy organelles occurs in a precise order to ensure that two viable daughter cells are generated post-cell division, and, in procyclic promastigote parasites, these events are accompanied by significant changes in cell morphology [2, 3] (S1 Fig).

At the start of the cell cycle (G1 phase), *Leishmania* promastigotes possess one nucleus (N), one kinetoplast (K) and one flagellum (F) (S1 Fig). During G1, the cell body doubles in length, but not in width [3]. The cell length then remains constant while DNA is replicated during S phase. About one hour before the end of S phase, basal body maturation occurs to nucleate the growth of a new flagellar axoneme, but the second flagellum only becomes visible by light microscopy when it emerges from the flagellar pocket around the end of S phase. Within the post-S phase period, G2 phase has not been precisely defined for *Leishmania*, and is assumed to be very brief [3]. Mitosis then ensues. G2 and mitosis are accompanied by a reduction in cell length and a simultaneous expansion in cell width. Reports differ as to the order of nucleus and kinetoplast division. In *L. mexicana*, until recently, the nucleus was thought to divide before the kinetoplast [3], while the reverse order has been reported in other *Leishmania* species [4, 5]. However, a recent study indicates these discrepancies may be down to the resolution of the imaging method, since 3D electron microscopy of dividing *L. mexicana* cells revealed that the final connection between the daughter nuclei (which may not be visible by fluorescence microscopy) is severed only after the completion of kinetoplast division [6]. Finally, cytokinesis, which involves localised remodelling of the microtubule cytoskeleton and invagination of the plasma membrane to form a cleavage fold and then a furrow to bisect the cell along the anterior-posterior (A-P) axis, completes the cell cycle [3, 5].

Thus, cell morphology and organelle replication are intricately linked in *Leishmania* procyclic promastigotes. Parasite cell cycle progression is commonly assessed through visualisation of organelles (through staining, immunofluorescence or the use of fluorescent marker proteins) alongside observation of cell morphology, as well as quantitation of DNA content *via* flow cytometry (typically using fixed cells stained with propidium iodide) [3, 7–9]. These methods are not without their limitations, though. While microscopy can provide high resolution spatial information, it can be highly labour-intensive if performed manually, for example, if rare populations of cells are being studied and many cells have to be screened to identify them, or if large numbers of cells are required for analysis or measurements of cell morphology

(e.g. length, width), which are usually performed by hand using software such as ImageJ/Fiji [3, 10]. Further, artefacts can be introduced if live cells are imaged for extended periods without the use of an environmental chamber, or if cells are fixed prior to staining. Conversely, flow cytometry approaches are high-throughput and can provide information on total DNA (or protein) content within a cell, but they do not offer information on how the DNA is packaged or the localisation of proteins or structures within the cell, and only provide limited information on cell size and morphology (e.g. from side scatter measurements).

Imaging flow cytometry (IFC) [11, 12] potentially offers a powerful alternative for cell cycle analysis, since it combines the high-throughput and quantitative analysis of flow cytometry with the visual and spatial information of microscopy by capturing images in brightfield, darkfield and fluorescence modes for every cell analysed. Further, the application of masks to identify areas of interest allows many morphological parameters, including length, width, perimeter, area, aspect ratio (AR) and circularity to be measured or calculated automatically. Upon identifying the morphological properties of a population of cells of interest (for example, short, wide, fluorescence positive or long, narrow, fluorescence negative), the application of masks and development of a gating strategy enables the automatic classification of cells matching these parameters. This has enabled the detection and morphological analysis of a wide range of cell types, from phytoplankton [13, 14] to blood cells [15–17], circulating endothelial and tumour cells [18–20], intracellular pathogens [21–23] and bacteria [24]. Further, IFC has been used to aid or automate cell cycle analysis of human cell lines and yeast, both with stained cell populations [25–28] and in unlabelled cells [26, 29], and can also increase the throughput and reproducibility of assays that detect subcellular phenotypes and activities, for example, DNA damage [30, 31], multinuclearity [32], changes to organelle structure or morphology [33, 34] and nucleocytoplasmic shuttling of proteins [35]. While IFC has previously been used to analyse several aspects of kinetoplastid biology [36–38], it has not, to our knowledge, been used to study their cell cycle.

Here we aimed to investigate if IFC could be applied to study the *Leishmania* cell cycle and whether it offered advantages over conventional methods of cell cycle analysis. We demonstrate that IFC offers a high throughput and convenient approach for analysing the *Leishmania* cell cycle in live cells. We find that Vybrant™ DyeCycle™ Orange quantitatively stains DNA in live promastigote cells and when combined with automated IFC measurements of cell body length and width, allows discrimination between early and late G1 phase cells, S phase, and G2/M/cytokinesis cells. Further, by tagging the orphan kinesin, KINF, with mNeonGreen, we use IFC to detail its cell cycle-dependent expression and localisation, and use the tagged cell line to provide greater resolution of cell cycle stages. For the first time, this allowed G2 cells to be distinguished from cells undergoing mitosis or cytokinesis, the durations of G2 and the post-mitotic period to be quantified, and newly divided G1 cells to be identified. Furthermore, IFC provided a rapid method to more precisely delineate the action of the CDK inhibitor, flavopiridol, previously reported to inhibit LmxCRK3 and affect the cell cycle at G2/M phase [39], and enabled an in-depth analysis of the timing and different routes of cell cleavage at the end of the cell cycle. The ability to capture tens of thousands of cell images within a few minutes, measure multiple morphological and fluorescence parameters for every cell, and then apply custom gates to automatically identify populations of interest from batches of data files, provides a step-change in the type and scale of analysis of *Leishmania*'s cell cycle and cell biology. Significantly, when applied to live cells, IFC offers near real-time analysis of perturbations to cell division within a population and offers the potential to determine the mechanism of action of novel drugs affecting the cell cycle.

## Results

### Imaging flow cytometry is suitable for morphological analysis of *L. mexicana*

To determine whether IFC is a suitable tool to analyse the *L. mexicana* promastigote cell cycle, live and paraformaldehyde (PFA)- or methanol-fixed C9T7 (parental) parasites were analysed by IFC to confirm that the previously reported cell cycle-dependent changes in cell morphology [3] could be observed (Fig 1). Cells with morphologies corresponding to all cell cycle stages were observed in the live and fixed populations, although some methanol-fixed cells appeared to have aberrant morphology (with cells appearing wider and more rounded) (Fig 1A). Measurements of length and width calculated using a cell body mask (Fig 1B and 1C) revealed that both PFA- and methanol-fixed cells were shorter in length (on average by around 8.8% and 38.1%, respectively) and wider (7.2% and 8.1%) than live cells (Fig 1D), indicating that fixing cells in solution alters their morphological parameters. For this study, therefore, only live cells were imaged.

### Vybrant™ DyeCycle™ Orange quantitatively stains DNA in live *L. mexicana* promastigotes

To determine the cell cycle position of individual *L. mexicana* parasites, a quantitative DNA stain suitable for use in live promastigote cells was required. Propidium iodide (PI) quantitatively stains DNA (S2A and S2B Fig), but requires cell fixation, and is thus unsuitable for use in live cells. Hoechst 33342 is commonly used to stain the nucleus and kinetoplast in live *Leishmania* [40–42], but IFC indicated that it does not bind proportionally to DNA under the conditions used here (S2C Fig). Images of Hoechst-stained cells revealed staining outside of the

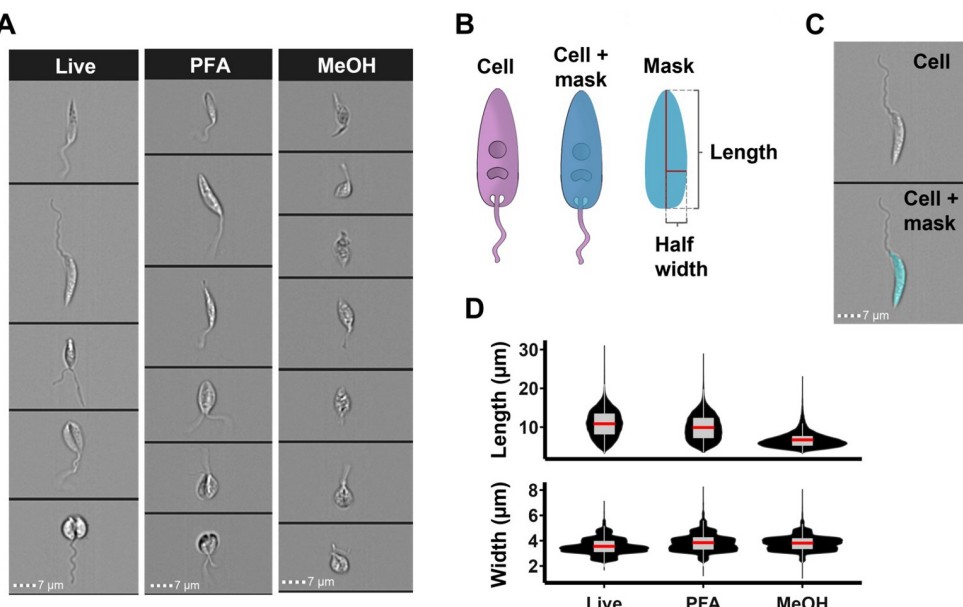

**Fig 1. Analysis of *L. mexicana* promastigote morphology using IFC.** (A) IFC images from live and PFA- or methanol (MeOH)-fixed promastigote *L. mexicana* C9T7 cells. (B) Schematic of cell body mask-based morphological measurements used for IFC analysis. (C) Images of a cell with and without a mask. (D) Violin plots of the length and width of cell populations from masked IFC images ($n > 26{,}499$). The box plots show the 25th and the 75th percentiles, and the mean in red. The whiskers represent the median +/- 1.5X interquartile range. Scale bars (dotted lines): 7 µm. See S1A Table for raw data.

nucleus that may account for it being non-quantitative (S2C Fig). DRAQ5 stained only the nucleus in the majority of cells and was not quantitative at concentrations of 5 μM or 50 μM, and, although potentially quantitative at 25 μM, resulted in alterations to cell morphology, suggesting it was toxic to the cells (S2D Fig). Vybrant™ DyeCycle™ Violet (DCV) stained both the nucleus and kinetoplast, but was not quantitative under the conditions tested (S2E Fig). Vybrant™ DyeCycle™ Ruby (DCR) was also non-quantitative as tested here, with staining appeared in foci throughout the cell body (S2F Fig). However, Vybrant™ DyeCycle™ Orange (DCO) was found to stain both the nucleus and kinetoplast throughout the cell cycle in C9T7 cells (Fig 2A), and to stain DNA quantitatively when cells were incubated in 0.625 μM DCO at room temperature for 30 mins (Fig 2B, 2C), analogous to PI in fixed cells (S2A and S2B Fig and Fig 2D). Note that consistent with previous fluorescence studies, the nucleus appears to divide before the kinetoplast [3]. Incubation with this concentration of DCO did not affect cell viability (S3 Fig), but a marked reduction in the fluorescence intensity of DCO-stained G2/M cells, which interfered with FCS Express™ modelling of the cell cycle peaks, was observed if there was a delay of >20 minutes in them being analysed by IFC following staining (S4 Fig). Hence, cells were analysed by IFC immediately after incubation with DCO. Other concentrations of DCO (from 1.25–10 μM) were also tested (at room temperature for 30 mins or at 27˚C for 10 mins) but they appeared to be toxic to the cells and/or to result in sub-optimal staining (S2G Fig).

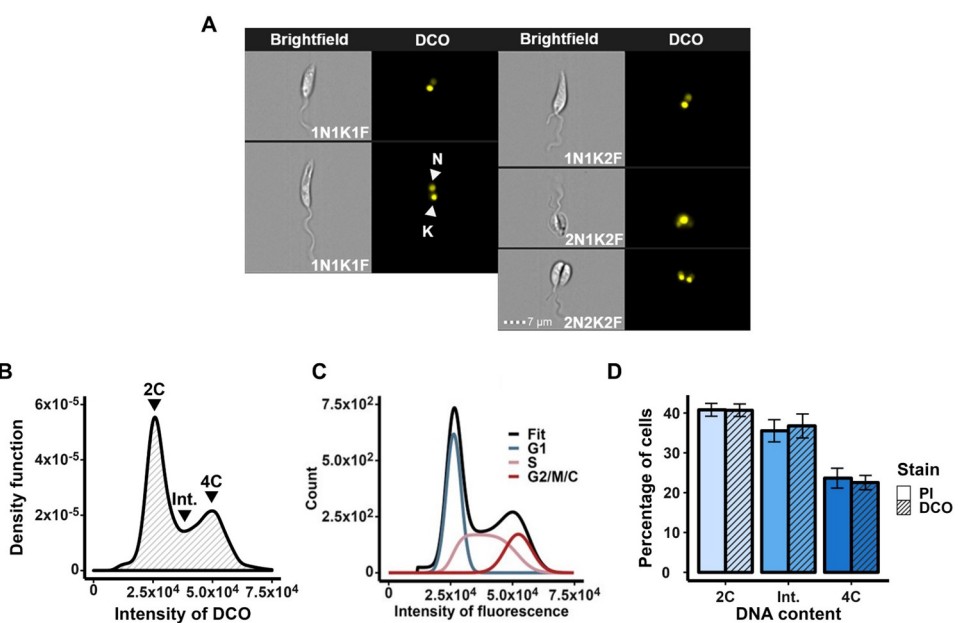

**Fig 2. DCO stains DNA quantitatively in live *L. mexicana* promastigotes.** (A) Brightfield and fluorescence images of *L. mexicana* promastigote C9T7 cells at different cell cycle stages stained with DCO. N: nucleus; K: kinetoplast; F: flagellum. Scale bars: 7 μm. Note that, consistent with previous fluorescence studies, the nucleus appears to divide before the kinetoplast. (B) The fluorescence intensity profile of C9T7 cells (*n* = 13920) stained with DCO and analysed with IFC. Peaks are indicated with arrowheads for non-replicated DNA (2C; corresponding to cells in G1 phase), replicated DNA (4C; cells in G2/M phase or cytokinesis) and cells with intermediate DNA content (Int.; cells undergoing S phase). (C) Cell cycle modelling of the DCO DNA intensity profile (panel B) using the FCS Express™ Multicycle engine (Rabinovitch & Bagwell debris subtraction [43, 44] and Dean/Jett/Fox cell cycle modelling [45]. The cell cycle stage of each curve is indicated. (D) Graph comparing the proportions of cells with different DNA contents, as modelled following staining with PI (S2A and S2B Fig) and DCO (panels B and C). Error bars represent the standard deviations of the means of three replicates. See S1B Table for raw data.

## Automatic cell cycle classification of promastigote cells using cell morphology

Due to biological variation and the cell cycle being a continuum, there is an overlap in the DNA contents of cells in G1 and early S phase, and of cells in late S and G2/M phases. To account for this natural overlap, cell cycle mathematical models such as the Dean/Jett/Fox model [45] (Figs 2C and S2B) or the Watson Pragmatic curve [46] can be used to extrapolate the proportions of cells in each cell cycle stage from DNA fluorescence measurements obtained by flow cytometry. Alternatively, manual gating can be performed where the user draws custom boundaries around the different DNA content peaks obtained [8, 47, 48]. Yet none of these methods consider the biology of the system nor can accurately classify the cell cycle stage of individual cells. Indeed, it has been previously reported that due to the long S phase, short G2 phase and rapid cytokinesis in *L. mexicana*, there is significant overlap of cell cycle stages within DNA intensity plots [49]. Given the significant morphological changes of *L. mexicana* cells during the cell cycle, and the ability of IFC to link morphological and DNA fluorescence data for individual cells, we hypothesised that IFC could result in greater resolution in classifying the cell cycle stage of individual cells than can be obtained using other methods such as microscopy and flow cytometry.

Live *L. mexicana* C9T7 promastigotes were stained with DCO and fluorescence intensity profiles were obtained using IFC. The 2C (G1 cells plus overlapping early S phase cells), intermediate (S phase cells plus overlapping G1 and early G2 phase cells) and 4C (cells in G2/M and cytokinesis, plus overlapping late S phase cells) DNA peaks were then manually gated (Fig 3A). Given the ability of IFC to analyse many cellular parameters, we asked which, if any,

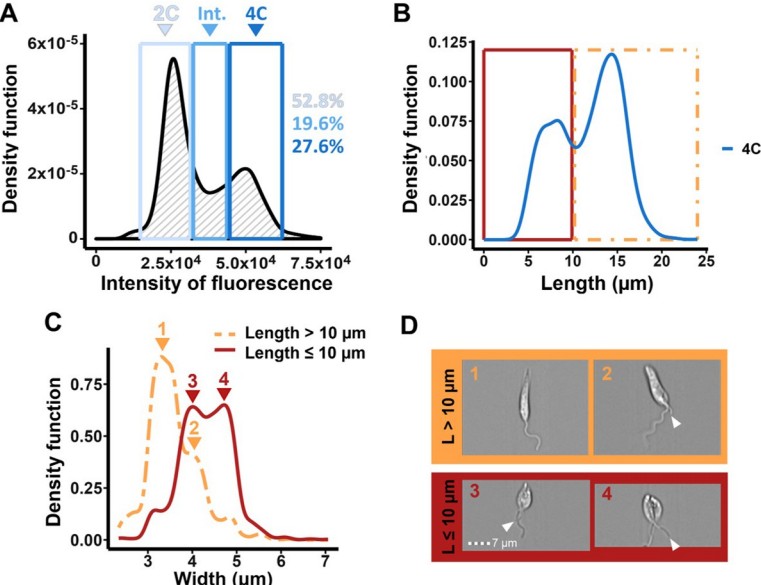

**Fig 3. Distinguishing late S, G2, M and cytokinesis cells based on their DNA content and cell body parameters.** *L. mexicana* C9T7 promastigote cells were stained with DCO and analysed by IFC. (A) The DNA fluorescence profile previously presented in Fig 2B was manually gated using IDEAS™ software (blue boxes) to select cells with 2C, intermediate (Int.; 2C-4C) or 4C DNA content based on the peaks. The percentages of cells in each peak are given on the right. (B) Plot of the cell body lengths of cells within the gated 4C population (blue line). A 10 μm cut-off was applied to separate the two peaks (red and yellow boxes). (C) Plot of the cell body widths of each of the two subpopulations identified in panel B (yellow (length > 10 μm) and red (length ≤ 10 μm), with example images from the respective peaks shown in (D). The distal end of the daughter flagellum in cells with two flagella is indicated by a white arrowhead. Scale bar: 7 μm.

morphological parameter(s) could allow cells in different cell cycle stages to be distinguished. Thirteen parameters (area, aspect ratio, brightfield intensity, circularity, compactness, diameter, elongatedness (height/width), height, length, perimeter, shape ratio, thickness_min and width) were plotted for cells present in each of the gates (Figs 3B and S5 and S6A). Of these, only elongatedness, height, length, perimeter and, to some extent, width offered discrimination between the cells within the three gates. Since the profiles for elongatedness, height, length and perimeter were very similar, we chose to proceed with cell body length, as this would allow the easiest comparison with previous morphological studies. Within the 2C gate, cell body lengths displayed a broad distribution, as expected, given that cells grow from their smallest length at the start of early G1, to their longest at the end of G1 phase, with the peak centred on 10 μm (S6A Fig). Within the intermediate (2C-4C) gate, the length distribution was shifted to the right, as expected for cells in S phase, peaking at ~13.5 μm, although there was a shoulder of shorter cells (~18% cells with cell bodies ≤10 μm in length), suggesting some overlap between G1 and S phases within this gate (S6A Fig). The width distributions for each of these gates were similar (S6B Fig), with the majority of cells < 4 μm in width, and width peaking at ~3.3 μm, consistent with previous observations that cells remain constant in width until G2 phase [3].

For the 4C gate, cell body length revealed a bimodal distribution, with peaks centred at ~7.5 μm and ~14.5 μm (Figs 3B and S6A). This is likely to reflect that late S phase and early G2 phase cells are long, but very rapidly shorten and widen during G2 phase and into mitosis [3]. Next, these peaks were gated to separate cells with cell bodies of ≤10 μm (36.6% cells; short) or >10 μm (63.4% cells; long) in length (Fig 3B). The range of morphological parameters described above were plotted for the long and short populations to determine whether any further discrimination of these populations could be achieved. Only aspect ratio, elongatedness and width offered a means of distinguishing the majority of short 4C-gated cells from all other cells gated on DNA content; none of the parameters selectively discriminated long 4C-gated cells (Figs 3C and S7). Again, to allow the easiest comparison with previous studies, we chose to focus on width.

Both long and short 4C-gated cell populations displayed a bimodal width distribution. Overall, the majority of the shorter 4C cells were wider than the longer 4C cells (Fig 3C). Based on previously described *L. mexicana* promastigote morphology [3], it is likely that the cells from the 4C gate with longer, narrower cell bodies comprised mostly late S/early G2 phase cells, while the 4C cells with shorter, wider cell bodies comprised mainly cells in late G2, M phase and cytokinesis. There was some overlap in width between the long and short cell populations (Fig 3C, subpeaks 2 and 3), most likely reflecting the changes to cell morphology that occur during G2 phase, as well as biological variation in cellular morphology. This was consistent with viewed images of the 4C cells corresponding to each subpeak (Fig 3D); long, narrow cells had either one flagellum (S phase), or one long and one very short flagellum (likely early G2 phase), while short, wide cells had two flagella (with a longer daughter flagellum) often accompanied by evidence of cytokinesis e.g. a division fold or cleavage furrow (G2/M/C cells). These data emphasise the pitfalls of relying on manual gating alone to determine cell cycle stage solely from DNA content, since the 4C gate includes a significant proportion of cells still in S phase in addition to post-S phase stages.

## Fluorescent tagging of KINF facilitates identification of cells in G2 and mitosis

Given the difficulties in unambiguously identifying the cell cycle stage of cells post-S phase from their DNA content and cell morphology alone, the spindle-associated orphan kinesin

KINF was tagged at its N terminus with mNeonGreen using CRISPR/Cas9 technology [50, 51] in *L. mexicana* promastigote C9T7 cells to allow the identification of cells undergoing mitosis. Analysis of cells expressing mNG:KINF by immunofluorescence using an anti-mNG antibody, in conjunction with the β-tubulin antibody, KMX, revealed cell cycle-dependent localisation (S8 Fig), consistent with a previous report of KINF localisation [52]. Cytoplasmic fluorescence was observed in all cells, but this was deemed to be background as similar fluorescence was observed in the C9T7 parental cell line (S8A Fig) and mNG:KINF was not detected in the cytoplasm when live cells were analysed by IFC (Fig 4A). In C9T7 mNG:KINF cells, mNG:KINF was expressed, as expected, in cells undergoing mitosis and shown to colocalise with the spindle microtubules (S8G and S8H Fig). However, following mitosis and disassembly of the spindle, mNG:KINF dispersed throughout each nucleus in 2N2K2F cells as daughter cells were separated by cytokinesis (S8I Fig). Amongst 1N1K1F cells, three groups of cells could be

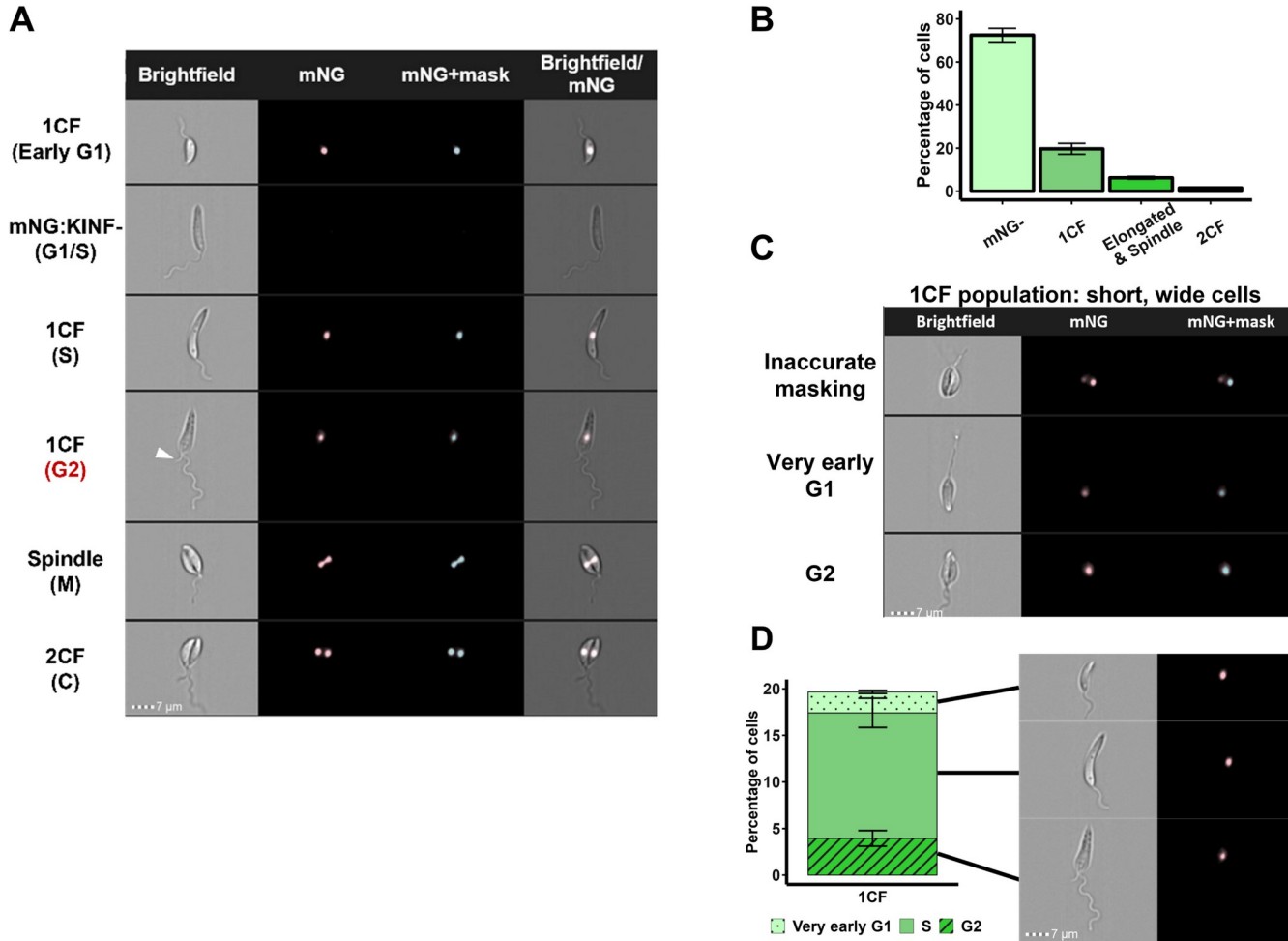

**Fig 4. Cell cycle-dependent expression and localisation of mNG:KINF facilitates the classification of cells in G2, M and cytokinesis.** (A) IFC images of live *L. mexicana* C9T7 mNG:KINF promastigote cells with differential expression of mNG:KINF (from left to right: brightfield, mNG:KINF (pink), IDEAS™-applied mask (cyan) and brightfield/mNG:KINF merge). The likely cell cycle stage of each example cell is indicated. A white arrowhead indicates the new flagellum that has just emerged from the flagella pocket in G2 phase. (B) Bar graph showing the proportions of cells displaying each expression/localisation pattern of mNG:KINF. (C) Examples of short and wide 1CF cells. Top: cell showing inaccurate masking of the mNG:KINF signal; such cells were subsequently reclassified as 2CF cells. Middle: cell in very early G1 phase (short, with 1F). Bottom: cell in G2 phase (with 2F). (D) % distribution of all cells within the 1CF population after the reclassification of the post-mitotic cells as 2CF. Example images of each cell type are shown. Scale bars: 7 μm. Error bars show standard deviation of the means of 3 replicates. See S1C–S1F Table for raw data.

distinguished based on their mNG:KINF fluorescence and cell body length. A few 1N1K1F cells with short cell bodies displayed mNG:KINF fluorescence distributed across their nucleus (S8B Fig) and it is likely that these were new G1 phase cells, with residual mNG:KINF fluorescence remaining after cytokinesis. Other 1N1K1F cells displayed no mNG:KINF fluorescence, including cells with short (S8C Fig) or long (S8D Fig) cell bodies, and likely represented cells further through G1 phase and into S phase. Additionally, some long 1N1K1F cells also displayed mNG:KINF fluorescence, as a single, roughly circular, focus, within the nucleus (S8E Fig) and likely represented cells later in S phase starting to synthesise mNG:KINF in preparation for mitosis. 1N1K2F cells without a visible mitotic spindle (G2 phase cells) also displayed mNG:KINF staining, distributed across the nucleus (S8F Fig). Thus, it appears that mNG:KINF expression commences partway through S phase, with the protein localising across the nucleus. mNG:KINF then localises to the mitotic spindle, and following spindle disassembly, redistributes across the nuclei, persisting into early G1 before, presumably, the protein is degraded. mNG:KINF staining therefore provides additional resolution of cell cycle progression compared to DNA staining alone, allowing the unambiguous identification of cells with a mitotic spindle as well as post-mitotic cells and cells in early G1.

## Use of mNG:KINF fluorescence to classify post-S phase cells

We then sought to determine if we could automatically classify and quantify the cell cycle expression pattern of mNG:KINF using IFC. In IFC images of C9T7 mNG:KINF cells, the mNG:KINF fluorescence signal was masked (Fig 4A) and the number of masks (and therefore discrete mNG:KINF signals) per cell was determined using the IDEAS™ spot count feature. This allowed cells with no, one or two mNG:KINF foci per cell to be distinguished. Further, cells with one mNG:KINF focus were expected, from immunofluorescence data to possess either a roughly circular focus in their nucleus or an elongated focus on the mitotic spindle (S8B, S8E–S8H Fig and Fig 4A), which could be distinguished by the aspect ratio intensity (ARI) of the mask. By visual inspection of cell images, cells with an elongated mNG:KINF signal (signifying mNG:KINF at the mitotic spindle) had an ARI < 0.62, while cells with a circular mNG:KINF focus in the nucleus had an ARI ≥ 0.62. Thus, it was possible to automatically classify C9T7 mNG:KINF cells as having a) no mNG:KINF foci (mNG:KINF-), b) one circular (nuclear) focus (1CF) of mNG:KINF, c) spindle-associated mNG:KINF (mitosis) or d) two circular foci (2CF) of mNG:KINF (post-mitosis) (S1C–S1F Table). These groupings were then classified into subgroups according to their cell body dimensions (length and width) to aid with cell cycle stage classification. At this stage, several anomalous subgroups were identified through manual visual inspection of the cell images from each group and were removed from further analysis (S1C–S1E Table). The largest of these subgroups comprised short, wide, mNG:KINF negative cells (~5.9% total cells) that were unclassifiable into any given cell cycle stage due to the mix of cells within this group; some cells had morphologies consistent with being in early G1, mitosis or cytokinesis but many had aberrant morphologies or appeared dead. Two subgroups of 2CF cells (short and narrow, and long and narrow, together accounting for ~0.9% total cells) contained a small extra-nuclear dot of mNG:KINF fluorescence in the region of the kinetoplast and were discarded due to the aberrant fluorescence pattern, and a further group of 2CF cells (long and wide; ~0.9% total cells), which were found to contain images with two cells side by side, were discarded because they had been misclassified by the automatic masking. Following removal of these aberrant cell groups, 72.4% cells were deemed to be mNG:KINF-, 19.7% had 1CF, 6.3% had spindle-shaped mNG:KINF fluorescence (mitosis) and 1.5% had 2CF mNG:KINF (post-mitosis) (Fig 4B and S1F Table).

Within the 1CF population, it was expected that there would be very early G1 phase cells with short, narrow cell bodies, S phase cells with long, narrow cell bodies as well as G2 cells with a range of morphologies, some of which would overlap with the morphologies of the other cell cycle stages. However, while these cell types were indeed observed during visual inspection of the 1CF cell images and associated cell body dimension scatter plots, an additional group of short, wide cells possessing two circular mNG:KINF foci was observed (0.3% total cells; Fig 4C and S1C–S1E Table). For these cells, only one of the two foci had been masked, likely because the second CF was in a different focal plane from the first (Fig 4C); these cells were reclassified as being post-mitosis. To identify G2 cells amongst the remaining 1CF cells (19.7% total cells), cell images were inspected by eye for the presence of two visible flagella (Fig 4C). 3.9% total cells were deemed to be in G2 phase by this criterion (Fig 4D and S1C–S1F Table) and, as predicted by previous analyses showing that post-S phase cells undergo significant morphological remodelling in preparation for mitosis [3], G2 cells displayed a wide variation in length and width (S9 Fig). Further, 1CF cells were judged to be in very early G1 if they were short and narrow with one flagellum (2.3% total cells) and in S phase if they were long with one flagellum (13.5% total cells) (Fig 4C and Tables 1 and S1C–S1F).

Inspection of cells classified as having mNG:KINF at the mitotic spindle, revealed a small group (average 0.23% total cells across three replicates) of long, narrow cells that only possessed one flagellum (S1C–S1E Table). From the orientation of the mNG:KINF fluorescence, which was aligned along the long axis of the cell, rather than perpendicular to it as is more normal during mitosis, it was judged that this subgroup was rather in S phase, with a slightly more elongated mNG:KINF signal than the majority of 1CF cells (S10 Fig); thus these cells were manually reclassified as S phase cells. From visual inspection, the 2CF cells contained only post-mitotic cells.

**Table 1. Properties of C9T7 mNG:KINF cell cycle stage populations.**

| Variable | Cell population | | | | | | | |
|---|---|---|---|---|---|---|---|---|
| Cell cycle stage | Very early G1 | Early G1 | Late G1 | Early S | Late S | G2 | M | Cytokinesis |
| NKF configuration | 1N1K1F | 1N1K1F | 1N1K1F | 1N1K1F | 1N1K1F | 1N1K2F | $1N^d1K2F$ | 2N2K2F |
| DNA content | 2C | 2C | 2C | ~2C-3C | ~3C-4C | 4C | 4C | 4C |
| Cell body length (µm) | ≤10 | ≤10 | >10 | >10 | >10 | any | ≤10 | ≤10 |
| Cell body width (µm) | <4 | <4 | <4 | <4 | <4 | any | ≥4 | ≥4 |
| mNG:KINF status | positive | negative | negative | negative | positive | positive | positive | positive |
| mNG:KINF pattern | 1 circular focus | n/a | n/a | n/a | 1 circular focus | 1 circular focus | spindle | 2 circular foci |
| % population | 2.3 | 17.7 | 27.9 | 27.0 | 13.4 | 3.9 | 6.1 | 1.7 |
| Duration of cell cycle stage | 12 mins (0.20 hrs) | 1 hr 40 mins (1.67 hrs) | 3 hrs 1 mins (3.02 hrs) | 3 hrs 30 mins (3.5 hrs) | 2 hr 2 mins (2.03 hrs) | 38 mins (0.64 hrs) | 1 hr 2 mins (1.03 hrs) | 19 mins (0.31 hrs) |
| Cumulative time through cell cycle | 12 mins (0.20 hrs) | 1 hr 52 mins (1.87 hrs) | 4 hrs 53 mins (4.88 hrs) | 8 hrs 23 mins (8.38 hrs) | 10 hrs 25 min (10.41 hrs) | 11 hrs 3 mins (11.05 hrs) | 12 hrs 5 mins (12.08 hrs) | 12 hrs 23 mins (12.39 hrs) |
| Duration in unit cell cycle stage | 0.016 U | 0.135 U | 0.243 U | 0.282 U | 0.164 U | 0.052 U | 0.083 U | 0.025 U |
| Cumulative time through unit cell cycle | 0.016 U | 0.151 U | 0.394 U | 0.677 U | 0.841 U | 0.892 U | 0.975 U | 1 U |

Based on mNG:KINF expression and localisation, cell morphology and DNA content, C9T7 mNG:KINF cells were classified into different cell cycle populations. The proportions of cells in each of the cell cycle populations was then used to calculate the duration of that stage. Cumulative time through the cell cycle was also calculated up to the end of each cell cycle stage using Williams' analysis [53]. Absolute timings were also converted to a proportion of the unit cell cycle, where cell cycle progress is scaled to between 0 and 1. N: nucleus; K: kinetoplast; F: flagellum; 2C, 3C and 4C: DNA content; U: units.

## Use of DCO staining to distinguish late G1 from S phase cells

Within the mNG:KINF- population, which comprises early and late G1 and early S phase cells, early G1 cells were identified through their short, narrow cell bodies (17.7% total cells). However, late G1 and early S phase cells (55.0% total cells) could not be separated on their morphological features alone because they both had similarly long cell bodies (Tables 1 and S1C–S1E and S1H). Therefore, we looked to distinguish them using DNA content. Upon analysis of mNG:KINF cells stained with DCO, it became apparent that this combination of fluorescence was unsuitable. Although mNG and DCO fluorescence emit in different channels of the imaging flow cytometer, they are both excited by a 488 laser and their optimal excitation was found to require very different laser powers (3 mW for DCO and 120 mW for mNG:KINF). Therefore, KINF was tagged with mCherry which is excited by a 561 nm laser, in an attempt to allow simultaneous determination of DNA content using DCO staining. However, the resultant mCherry:KINF fluorescent protein was not bright enough to be detected by IFC. Instead, a second aliquot of the C9T7 mNG:KINF cells analysed for mNG:KINF fluorescence was stained with DCO and excited with the 488 nm laser at 3 mW. This permitted the detection of DCO staining in C9T7 mNG:KINF cells without exciting detectable mNG:KINF fluorescence (S11A Fig), enabling the quantification of cellular DNA content by applying manual 2C, intermediate (2C-4C gate) and 4C gates (see example plot in S11B Fig and data for three replicates in S1G and S1H Table). Given the overlap of G1 and S phase cells within the 2C and intermediate gates (S6A and S6B Fig), a precise determination of the proportions of late G1 and early S phase cells was not possible. However, an estimate of late G1 phase cells was made by subtracting the proportion of the short ($\leq$ 10 μm) cells with 1 CF mNG:KINF fluorescence (very early G1 phase; 2.25%) or no mNG:KINF fluorescence (early G1 phase; 17.65%) from the average percentage of 2C cells quantified by DCO staining (47.8%, *n = 3*). Thus, 27.9% cells were estimated to be in late G1 phase. Subtracting this value from the 55.0% of long mNG:KINF- cells (late G1 and early S phase cells combined) provided an estimate of 27% total non-fluorescent cells being in S phase (Tables 1 and S1H).

The resultant percentages allowed the durations of the various cell cycle stages to be calculated [3, 53] for the C9T7 mNG:KINF cell line based on its doubling time of 12 hours 23 minutes (12.4 hours) (Table 1). Cell cycle timings were also converted to a fraction of the unit cell cycle (where progress through the cell cycle is scaled to between 0 and 1). This showed that G1 phase lasted for ~4.9 hours, S phase for ~5.5 hours, G2 phase for just 38 minutes (0.64 hrs), mitosis for ~1 hr and the post-mitotic period for 19 minutes (0.31 hrs), equating to 0.4, 0.45, 0.05, 0.08 and 0.03 of the unit cell cycle, respectively, consistent with the unit cell cycle values calculated by Wheeler et al, 2011 [3]. This is the first determination of the length of G2 and the post-mitotic period in *L. mexicana*. The cell cycle timings also revealed that mNG:KINF expression commenced ~60% of the way through S phase (8.4 hrs or 0.68 U into the cell cycle) with mNG:KINF being present for the remainder of the cell cycle before being degraded very early (12 minutes) into the new cell cycle.

## Using IFC images to investigate cytokinesis

We next endeavoured to provide a more precise timing for cytokinesis, by observing dividing cells within the mitotic and post-mitotic populations. The first stage of cytokinesis detectable by light microscopy is cleavage fold formation where the plasma membrane and the underlying subpellicular microtubules begin to fold inwards in the centre of the cell along the A-P axis. This is followed by cleavage furrow ingression where the microtubules and plasma membrane are split to give rise to two daughter cells. A recent report documents that furrow ingression in *L. mexicana* occurs unidirectionally, from the anterior to posterior [6], analogous to its

close relative, *Trypanosoma brucei* [54]. However, an earlier study indicated that cytokinesis in *L. major* could also occur from the posterior to anterior [5].

Visual inspection of images of post-S phase cells with their A-P axis visible identified the formation of cleavage folds in some late G2 cells, and in virtually all cells with spindles (Fig 5B), suggesting that cells start cytokinesis around the G2/M transition. However, ingression of the cleavage furrow was only observed in post-G2 phase cells. Cleavage furrows were observed in ~53% cells with mitotic spindles, rising to ~78% cells with 2CF mNG:KINF, suggesting variability in the timing of the initiation of furrowing, with some cells commencing furrowing during mitosis, and others not starting to furrow until after completion of mitosis (Fig 5A and 5B and S1I Table). Further, we observed considerable variability in the directionality of furrowing (Fig 5A and S1I Table), with four routes to cell cleavage observed: i) unidirectional furrowing from anterior to posterior (Fig 5C), ii) unidirectional furrowing from posterior to anterior (Fig 5D), iii) bidirectional furrowing from each pole (Fig 5E) and, less frequently, iv) furrowing commencing internally (rather than at a pole) (Fig 5F). In cells still undergoing mitosis, furrow ingression was observed most commonly to begin from the posterior cell pole (Fig 5A). However, in post-mitotic cells, approximately equal proportions of cells were observed with anterior, posterior or bidirectional furrows (Fig 5A and S1I Table), suggesting that there was no particular preference for any one of these routes. For each route, cells with cleavage furrows that had ingressed almost the full length of the cell were observed, suggesting that each route of furrowing could progress to completion. Furthermore, a small population (4–6%) of post G2-phase cells (spindle or 2CF cells) appeared to commence cell cleavage internally, close to the posterior pole, while the daughter cells were apparently still joined at their posterior tips, with the furrow then seeming to extend internally along the A-P axis, highlighting the flexibility of cytokinesis in *L. mexicana*. Thus, cytokinesis appears to take the whole mitotic and post-mitotic period to complete (~1 hr 20 mins).

Taken together, our data provide a more detailed set of timings of the different cell cycle stages in promastigote *L. mexicana* as shown in the scaled schematic in Fig 6.

## Investigating the action of flavopiridol

We hypothesised that our developed IFC pipeline and mNG:KINF cell line could be used to investigate the mechanism of action of novel drugs that affect the cell cycle. To provide proof-of-principle for such an application, the effects of the cell cycle inhibitor, flavopiridol were investigated. Flavopiridol is an ATP-competitive CDK inhibitor that has been reported to inhibit CRK3 in *L. mexicana*, resulting in cells arresting at G2/M phase, but the exact point of cell cycle arrest has not been previously determined [39]. Here, *L. mexicana* promastigote mNG:KINF cells were incubated in 5 μM flavopiridol for 12.4 hours (equivalent to one cell cycle) before being analysed by IFC. PI staining of the treated cells revealed that incubation with flavopiridol resulted in a 3.5 fold enrichment (from 22% to 74%) of cells with 4C DNA content and a concomitant decrease in cells with 2C and intermediate DNA content compared to control cells (Fig 7A and S1J Table), indicating that cells were arrested at the end of S phase or beyond. Analysis of mNG:KINF fluorescence in flavopiridol-treated cells revealed that mNG- cells decreased by two thirds (57% to 20%) while cells with 1CF of mNG:KINF increased nearly two-fold (from 38% to 74%) (Fig 7B and S1J Table) compared to untreated cells. In contrast, there was little change to the proportion of cells with spindles or 2CF mNG:KINF, indicating that flavopiridol did not noticeably arrest cells in mitosis or beyond. Examination of the images of the 1CF cells in flavopiridol-treated cells revealed that the majority (>80%, *n* >110) possessed two flagella (Fig 7C and S1J Table), indicating that they were in G2 phase, rather than at the end of S phase.

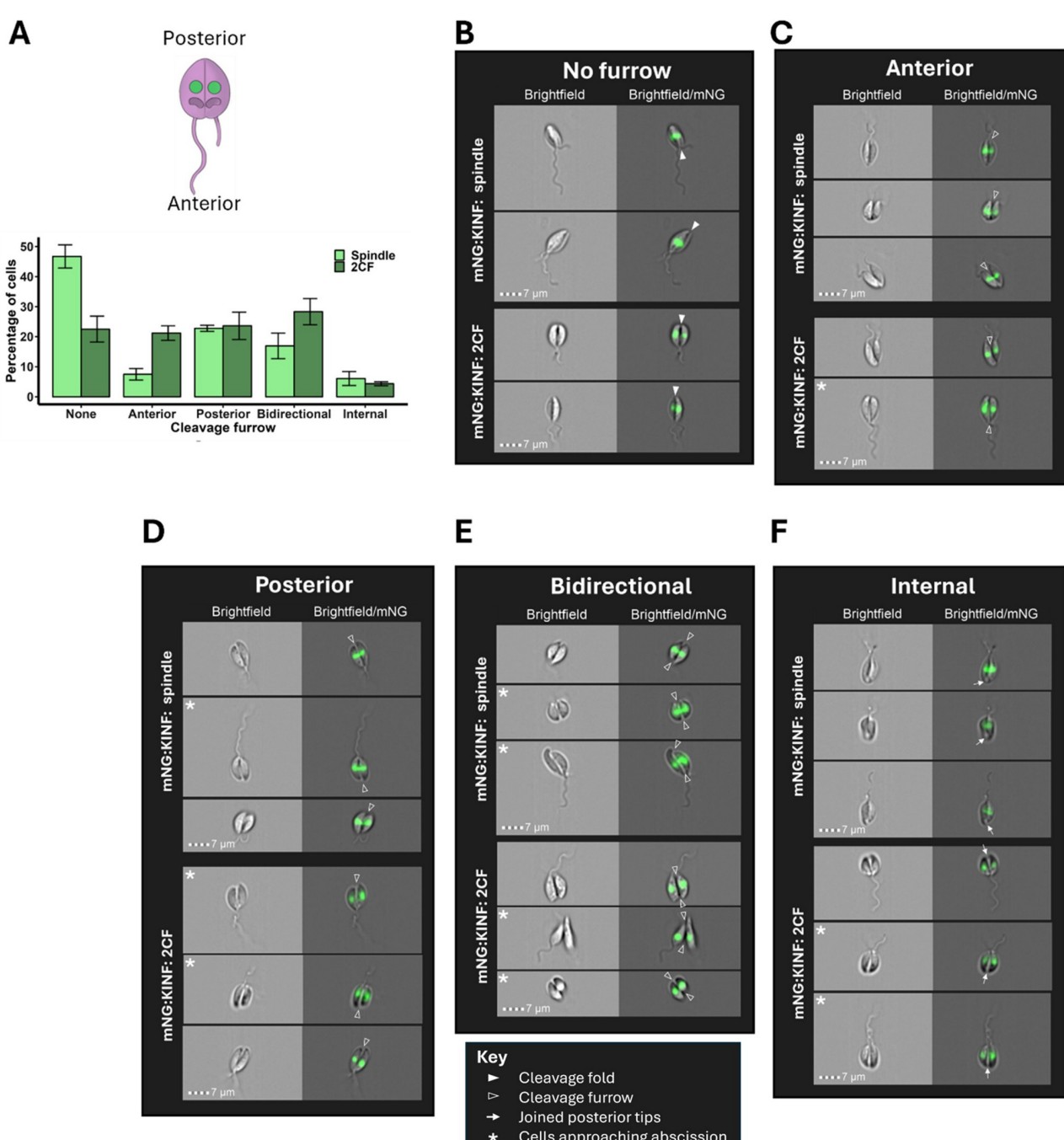

**Fig 5. Cytokinesis is highly pleiomorphic in _L. mexicana_ promastigotes.** (A) Images of mitotic and post-mitotic C9T7 mNG:KINF cells, as determined from their mNG:KINF fluorescence pattern (spindle and 2CF), were scored according to their cytokinesis stage. Cell images were only included in this analysis if the cell's A-P axis was visible; cells imaged from their poles were excluded. The cartoon indicates the anterior and posterior poles of a _L. mexicana_ cell. Graph summarises the data from 3 replicate C9T7 mNG:KINF cultures and 146–197 cell images for each spindle and 2CF population. Error bars indicate the standard deviation of the mean. See S1I Table for raw data. (B-F) Example images of spindle and 2CF C9T7 mNG: KINF cells demonstrating different modes of cytokinesis (B: cleavage fold present but no furrow; C: furrowing from anterior pole; D: furrowing from posterior pole; E: furrowing from both poles; F: furrowing commencing internally along the A-P axis). Brightfield and brightfield/mNG:KINF merged images are shown. Closed arrowheads: cleavage fold; open arrowheads: cleavage furrow; arrows: joined posterior cell tips beyond the internal cleavage furrow; asterisks: cells where cleavage furrow is almost complete and cells are approaching abscission. Scale bars: 7 μm.

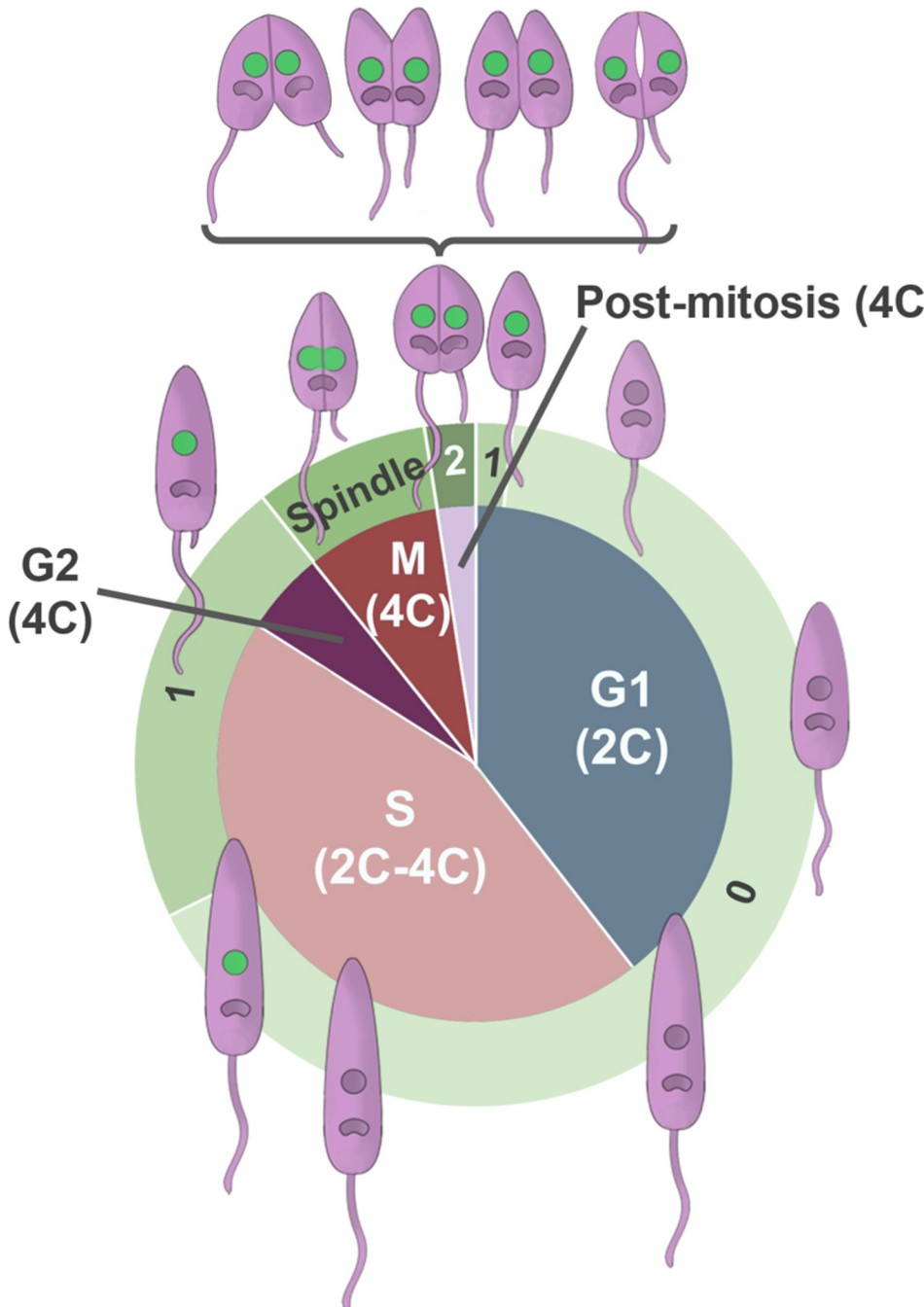

**Fig 6. Schematic of the *L. mexicana* cell cycle.** The inner circle shows the five main stages of the cell cycle, scaled according to their duration, along with the DNA content of cells progressing through those stages. The outer circle indicates the cell cycle-dependent expression of mNG:KINF, localising either to circular foci within the nucleus (1 or 2) or to the mitotic spindle. The '0' indicates cells with no mNG:KINF expression. Cartoons of cells at each cell cycle stage showing the cell body and flagella (light purple), nuclei and kinetoplasts (dark purple), and cleavage fold are indicated on the outside. The bracketed cells indicate the diversity of furrowing observed in both mitotic and post-mitotic cells (anterior to posterior, posterior to anterior, bidirectional and internal). mNG:KINF nuclear fluorescence is in green.

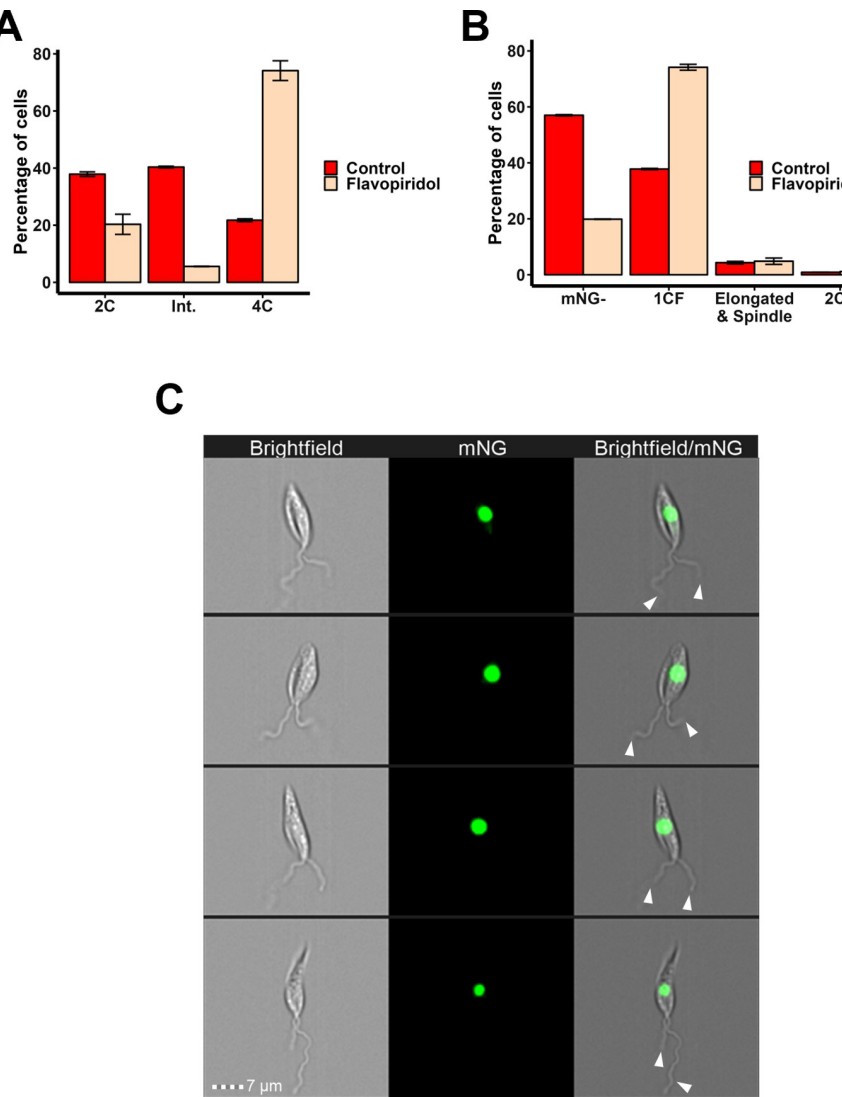

**Fig 7. Flavopiridol arrests cells in G2 phase.** (A) *L. mexicana* promastigote C9T7 mNG:KINF cultures were incubated with flavopiridol or not (control) for one cell cycle (12.4 hrs) before being fixed in MeOH, stained with PI and analysed by flow cytometry. DNA content of individual cells ($n > 10,000$ per replicate) from duplicate cultures was assessed using FCS Express software and plotted (error bars: standard deviation). (B) The same cultures as in (A) were also analysed live by IFC (unstained), using the image analysis pipeline described above to determine the fluorescence pattern of mNG:KINF. $n > 6500$. (C) Example IFC images of 1CF cells with two flagella (white arrow heads) that accumulated following flavopiridol treatment. Brightfield, mNG and merged images are shown. Scale bar: 7 μm. Raw data is in S1J Table.

## Discussion

Much of our current knowledge of *Leishmania* cell cycle events and their order and duration derives from studies where fixed promastigote parasites have been analysed manually by fluorescence microscopy [3, 5, 8, 10, 55]. DNA dyes are commonly used to visualise and enumerate nuclei and kinetoplasts, or thymidine analogues such as BrdU (5-bromo-2'-deoxyuridine) or EdU (5-ethynyl-2'-deoxyuridine) are used to detect DNA synthesis. Immunofluorescence of cytoskeletal proteins may aid in visualising and documenting organelle replication, and cell morphology is generally assessed under phase contrast and may be supplemented with

electron microscopy to analyse the structural detail of cells. Cell morphological parameters such as cell body or flagellum length and width may be measured and DNA content measurements performed on individual cells using software such as ImageJ [56, 57]. While there is no doubt that such studies are often very detailed and carefully performed, the laborious nature of manual analyses have limited the numbers of cells able to be analysed to a few hundred, at most, per replicate [3, 5]. Often, microscopy is supplemented with flow cytometry analysis of total cellular DNA content of PI-stained cells [8], which, while offering greater throughput (tens of thousands of cells analysed per replicate), does not provide data on how many organelles are in each cell, and only limited information about cell size and shape.

High throughput analysis of microscopy images (of fixed or live cells, stained with DNA dyes and/or expressing fluorescently tagged proteins) has also been developed for kinetoplastids, using ImageJ scripts to automate measurement of DNA content and nucleus and kinetoplast number per cell, cell length, width, shape and the position of nuclei and kinetoplasts along the long axis of the cell [49, 58]. Further, the nuclei and kinetoplasts in *L. mexicana* and the related kinetoplastid, *Trypanosoma brucei*, can be differentially stained with different DNA dyes [49, 59]. Base-pair intercalators (BPIs) such as PI and SybrGreen, do not have any particular sequence specificity and so stain the nucleus, which contains more DNA, more brightly than the kinetoplast. In contrast, minor groove binding dyes (MGBs), such as DAPI (4‘, 6-dia-midino-2-phenylindole) and Hoechst 33342, have a greater affinity for A-T-rich DNA and therefore preferentially stain kinetoplast DNA over the nuclear DNA. Co-staining with one BPI and one MGB dye allows, following chromatic aberration correction and colour deconvolution of acquired fluorescent images, the automated detection of nuclei and kinetoplasts, and the separate measurement of their DNA content using ImageJ [49]. However, this approach requires that cells are imaged on microscope slides prior to ImageJ analysis, with both sample/slide preparation and image capture being critical for accuracy and consistency and, potentially, rate-limiting.

Here, we have developed new cell cycle analysis pipelines for live *L. mexicana* promastigotes in solution that require only minimal manipulation (washing, staining (if required) and resuspension in PBS) of the parasites prior to their analysis by IFC. Imaging cells live and in solution offers several advantages over imaging cells on microscope slides. Fixation artefacts are not introduced, cells cannot dry out and cell morphology is not altered by coming into contact with a surface. The throughput of IFC greatly surpasses that of microscopy, with tens of thousands of cell images and dozens of parameters for each image being captured in as little as 5 minutes. While the resolution achieved by IFC (numerical aperture of 0.9) is lower than that of fluorescence microscopy (numerical aperture of ~1.4 for many 63x or 100x objective lenses), and there is a need to gate out any out-of-focus cells prior to image analysis, the pixel size (0.9 $\mu m^2$) and resultant image quality is nevertheless sufficient to clearly visualise the single copy organelles of *Leishmania*. Being able to view brightfield and fluorescence images of every cell analysed, and to automatically quantify their many parameters without needing to write custom scripts, offers hereto unparalleled insights into the *Leishmania* cell cycle, and provides far greater consistency of analysis than has been possible to date with other methods. Additionally, because IFC captures all possible brightfield and fluorescent parameters for every cell image at the outset, this permits backtracking during analysis to explore additional parameters for any given cell, even if these were not part of the initial experimental design, as well as to visually verify gated cells, and, as we have done here, to supplement automatically acquired data with manual analysis, thereby providing enormous flexibility.

To be able to take advantage of all that IFC has to offer, we first had to find a DNA dye that was quantitative in live *L. mexicana* cells. The BPI, PI, is quantitative (S2A and S2B Fig), but can only be used in fixed, RNAse-treated cells due to it being non-membrane permeable and

also binding to RNA. Despite the MGB Hoechst 33342 being widely used to stain DNA in live *Leishmania*, it was not quantitative under the conditions used here (S2C Fig), which was surprising because Hoechst is a quantitative stain for live *T. brucei*. However, examination of images of cells stained with Hoechst showed that they also displayed regions with extra-nuclear fluorescence, suggesting that Hoechst staining was not restricted to the nucleus. The BPI, DRAQ5, in contrast, appeared visually to label only the nucleus (S2D Fig), but was either not quantitative or caused toxic effects under the conditions used. While the DNA binding mechanisms of the Vybrant Dye Cycles are to our knowledge, not known, DCV appeared to specifically label both the nucleus and kinetoplast, with brighter staining of the kinetoplast, suggesting it might be an MGB. However, it was also not quantitative (S2E Fig), despite being so in *T. brucei* bloodstream form cells [60]. DCR also was not quantitative, staining both the nucleus and kinetoplast, but also throughout the cytoplasm (S2F Fig) under the conditions tested. However, we found a third Vybrant DyeCycle dye, DCO, previously reported to be quantitative in procyclic for *T. brucei* [61], was quantitative at a much lower concentration (0.625 μM) in promastigote *L. mexicana* than that used for *T. brucei* (5 μM), provided cells were imaged promptly after staining (Figs 2 and S4). We also showed it to be non-toxic at this concentration (S3 Fig), which is important given that DCO, along with DRAQ5 and Hoechst 33342, were reported to display cytotoxicity and cause G2/M arrest in HeLa cells [62, 63]. DCO stained both DNA-containing organelles, preferentially labelling the kinetoplast, suggesting it may be an MGB. Since S phase of the nucleus and kinetoplast are synchronous in *L. mexicana* promastigotes, this dye worked well to determine relative DNA content of cells throughout the cell cycle, providing comparable data to PI (Figs 2 and S2A and S2B), although the preferential binding to kDNA should be borne in mind in the event of analysing DNA content under conditions where DNA synthesis of one of these organelles is preferentially affected.

Staining live *Leishmania* with DCO before imaging with IFC not only allowed a cell's DNA content to be determined, giving an approximation of its cell cycle position (in a manner comparable to staining fixed cells with PI), but also allowed DNA content to be matched to cell morphology, allowing greater level of automatic discrimination of some cell cycle stages, e.g., cells in G1 phase (2C gate) could be split into early and late G1 cells on the basis of their length (Table 2). However, distinguishing late S phase cells from early G2 cells, or late G2 cells from mitotic cells was not possible from their DNA content and cell body length/width due to their identical DNA content and overlapping dimensions. To delineate the start and end of G2 phase, we employed fluorescent labelling of the orphan spindle kinesin, KINF, to detect spindle formation at the start of mitosis and disassembly at the end of mitosis. Serendipitously, we found that mNG:KINF expression and localisation allowed us to also identify mid-S phase-G2 cells (one circular focus, long cells), post-mitotic cells in the latter stages of cytokinesis (2 circular foci in short, wide cells) and newly divided cells (one circular focus, short, narrow cells) (see summary in Table 2). We were then able to distinguish late S phase and early G2 phase cells by examining images of these cells to determine whether a second daughter flagellum had emerged from the flagellar pocket. Although we performed this manually in this study, we believe this should be possible to automate by, for example, fluorescently tagging the axoneme capping structure protein (ACS1; [64]) at the distal tip of the flagellum with a fluorescent tag such as mCherry, mScarlet3 [65] or cyan fluorescent protein [66] and using the IDEAS™ spot count feature to distinguish between one and two ACS1 foci.

One other issue we encountered using DCO and mNG:KINF for cell cycle analysis was that, despite their fluorescence emitting in different channels, they both require a 488 laser for excitation but with different laser powers for optimal fluorescence, meaning that duplicate samples had to be analysed in tandem, rather than the DCO and mNG:KINF signals being

**Table 2. Summary of cell cycle data attainable using IFC pipelines.**

| Analysis parameter | | | Attainable data | | | |
|---|---|---|---|---|---|---|
| Cell morphology | DCO staining | mNG:KINF fluorescence | DNA content & NK number | Flagellum number | Spindle detection | Cell cycle periods resolvable |
| √ | | | No | Yes | No | Early G1; late G1-S; early G2; late G2-post-M; furrowing |
| | √ | | Yes | No | No | G1-early S; S; late S-M; post-M |
| | | √ | No | No | Yes | Very early G1; G1-S; late S-G2; M; post-M |
| √ | √ | | Yes | Yes | No | Early G1; late G1-early S; S; early G2; late G2-M; post-M; furrowing |
| √ | | √ | No | Yes | Yes | V. early G1; early G1; late G1-S; late S; G2; M; post-M; furrowing |
| √ | √ | √ | Yes | Yes | Yes | Very early G1; early G1; late G1; early S; late S; G2; M; post-M; furrowing |

Summary of the possible cell cycle outputs attainable using the IFC pipelines developed in this study following analysis of cell morphology, DCO staining and mNG: KINF fluorescence alone and in combination. N: nuclei; K: kinetoplasts; G1 and G2: gap phases; S: DNA synthesis; M: mitosis.

analysed simultaneously in a single sample. Efforts to swap the fluorescent tag on KINF to mCherry to render it compatible with DCO were thwarted by the low fluorescence of mCherry:KINF (apparent even by fluorescence microscopy), that rendered it undetectable by IFC. Our data do not necessarily preclude mCherry being useful for IFC; indeed, IFC has been used to track host-parasite interactions using mCherry-expressing *Toxoplasma* parasites [67], but for KINF, a different fluorescent tag could be trialled e.g. the recently developed mScarlet3, which is more than five times brighter than mCherry [65], or perhaps a cyan fluorescent protein [66] to find an alternative that also permits co-staining and analysis with DCO.

Being able to distinguish cells in different cell cycle stages allowed the calculation of their duration and allowed us to determine the length of the G2, mitosis and post-mitosis phases separately for the first time (Table 1). While the doubling time of our C9T7 mNG:KINF cell line (12. 4 hrs) is considerably longer than that of the wildtype WHO strain MNYC/BZ/62/ M379 cell line (7.1 hrs) used by Wheeler *et al*., [3], the unit cell cycle durations of the pre-S (0.40 U), S (0.45 U) and post-S (0.16) phases we determined are highly consistent with the equivalent ranges reported by Wheeler *et al*., from four different analysis methods (pre-S: 0.35–0.4; S: 0.4–0.49; post-S: 0.14–0.22). Further, we provide much greater resolution of cell cycle stages, being able to distinguish three stages of G1 phase (newly divided cells (v. early G1), early and late G1), early and late S phase, G2 phase, mitosis and post-mitosis (Table 1). Of particular note is the ability, because of the high-throughput nature of IFC, to detect reliably cells in short-lived cell cycle stages, for example, new cells within the first 12 minutes of the cell cycle (2.3% cells), G2 phase (38 minutes; 3.9% cells) and post-mitotic cells (19 minutes; 1.7% cells). In the case of the latter, 479 post-mitotic cells across three replicates from the 2CF gated population were visually inspected for the presence of a furrow. If such analysis had been carried out manually on unsorted populations (e.g., by microscopy), >28,000 cells would have had to have been viewed in order to have identified this number of 2CF cells for analysis. Clearly, to manually inspect such large numbers of cells is unfeasible, demonstrating the benefits IFC offers for the identification and analysis of rare cell populations. Further, the ability to view substantial numbers of cells from rare cell cycle populations could negate the need to use enrichment techniques such as cell cycle synchronisation with agents such as hydroxyurea or flavopiridol [8, 39, 68, 69] that may introduce artefacts.

Here, the high throughput nature of IFC allowed us to gain new insights into cytokinesis in *L. mexicana* promastigotes (Fig 5 and S1I Table), showing that the cleavage fold formed at the G2/M transition, while initiation of furrowing was more variable, with some cells commencing furrowing during mitosis and others not starting until mitosis was complete. Furthermore, cleavage furrows were observed to ingress unidirectionally from the anterior or posterior of the cell, bidirectionally from both poles simultaneously, or to initiate internally along the A-P axis. This suggests that while cytokinesis is constrained to occur at the midpoint of the cell along the A-P axis, following the inward folding of the plasma membrane and subpellicular microtubules, cleavage of the membrane and microtubules can initiate at many points along the axis. Thus, cytokinesis in promastigote *L. mexicana* differs considerably from that in the related kinetoplastid, *T. brucei* [54, 70], and may reflect that *Leishmania* cells round up before dividing, altering the membrane curvature which could influence furrow ingression [71, 72]. Such flexibility in cytokinesis has been reported before in *L. major* [5], but not remarked on in other studies [3, 6], which may be a reflection of the lower number of dividing cells analysed (<150 post-S phase cells in [3] and just four furrowing cells in [6]). Alternatively, since adhesion to a substratum provides traction forces and has been shown to influence the mode of cytokinesis in *Dictyostelium* [73] and animal cells [74], it is tempting to speculate that the increased modes of cytokinesis visualised here may reflect that cells were analysed in solution and thus able to move freely without attaching to a substratum, rather than with their movement constrained on microscope slides or in agarose, which might preferentially favour furrowing from anterior to posterior.

Finally, we provide proof of principle, using the CDK inhibitor, flavopiridol, that the cell cycle analysis pipelines we have developed here can also be used to identify the effects of drugs on the cell cycle. Flavopiridol was reported over 20 years ago to inhibit the CDK, LmxCRK3, and enrich for *L. mexicana* cells with a 4C DNA content but only one nucleus and one kinetoplast, indicating that the cell cycle was arrested at G2 or in mitosis, before nucleus or kinetoplast division [39]. Here, using IFC and mNG:KINF cells, we narrow its action to G2 phase, suggesting that the activity of its target, LmxCRK3, is required to signal mitotic spindle formation, and envisage that our pipeline could be used to determine the action of any compound or environmental condition that affects the cell cycle.

Overall, our application of IFC to studying the cell cycle in *Leishmania* provides an extra dimension to the growing body of IFC applications for studying the cell biology of parasites. IFC has been used to assess morphological changes occurring during promastigote to amastigote differentiation in *L. donovani* [75], to assess colocalisation of mCLING and lysotracker fluorescent stains in *T. brucei* [36] and to assess PI live/dead staining and for morphological measurements in *L. mexicana* [38]. IFC has also been employed extensively to study host-parasite interactions for *Leishmania* and other intracellular pathogens [21, 76–79] as well as to detect *Leishmania* infections in blood and parasite dissemination to organs around the body [80]. It has also been used to study trogocytosis by *Entamoeba histolytica* [81] and for drug screening in *Plasmodium falciparum* [82]. The huge flexibility and high-throughput nature of IFC provides opportunities to study the cell biology of parasites such as *Leishmania* at a scale and level of detail that hitherto has not been possible. The cell cycle analysis pipelines we have developed here should, for example, provide the means to identify and extensively characterise cell cycle and morphological perturbations in mutant or drug-treated cell lines, to study the expression and localisation of proteins of interest throughout the cell cycle and to identify rare phenotypes that only occur in a small subset of cells. Furthermore, IFC provides a simple method of automated image capture for live cells, reducing artefacts and experimental variation, as well as standardised and largely automated image analysis that allows the visual

insights of microscopy to be linked with the high-throughput quantitation of flow cytometry for every individual cell, providing a step change for *Leishmania* cell cycle analysis.

Further improvement of our IFC cell cycle analysis pipeline might be achieved in future by using machine learning or neural networks to automatically classify cell images, thereby simplifying the gating strategies required. Machine learning and/or neural networks, trained using a ground truth dataset from manually classified cells, have been used to automatically classify different life cycle stages of phytoplankton [14], different species of pollen [83], the presence of *Giardia* and *Cryptosporidium* parasites in drinking water [84] and different mammalian cell cycle stages [29, 85], using brightfield (or brightfield and darkfield) image training sets. We attempted to use the IDEAS® Machine Learning module to achieve automatic cell cycle classification of brightfield images of *L. mexicana* promastigotes stained with Hoechst, using a training set of brightfield images which had been manually classified into different cell cycle stages according to their corresponding DNA fluorescence images. However, the module was unable to accurately identify different cell cycle stages, for example failing to differentiate between cells with one or two flagella. Employing a larger training dataset, or using one based on fluorescence images might yield greater success in the future, and it would be of interest to assess whether deep learning software such as Amnis® AI Image Analysis Software could provide additional capability for *Leishmania* cell cycle analysis.

## Materials and methods

### *Leishmania* cell culture

*Leishmania mexicana* M379 Cas9 T7 (C9T7; [51]) promastigote cells were cultured in medium 199 (M199; [50]) supplemented with 10% foetal bovine serum (FBS), 32 μgml$^{-1}$ hygromycin and 50 μgml$^{-1}$ nourseothricin sulphate at 27˚C. All analyses were conducted using cells grown to mid-log phase (1x10$^6$ – 7x10$^6$ cellsml$^{-1}$).

### Generation of an mNG:KINF tagged cell line

The spindle-associated orphan kinesin F (KINF; LmxM.25.1950) was tagged with mNeon-Green (mNG) at its N-terminus using CRISPR-Cas9 technology [50]. PCR was performed to generate the sgRNA and DNA donor templates with primers designed using LeishGEdit [86] [upstream forward primer: 5`-TCGTTCTCTCCGTCTCTCGGGGTGGCTGCCgtataa tgcagacctgctgc-3`, upstream reverse primer: 5`-TACCGCCACCTTGATGCTAGACGA CGACATactacccgatcctgatccag-3` and 5' sgRNA primer: 5`-aaattaatacgact cactataggCTCCTTGCACCAGCAAAGACgttttagagctagaaatagc-3`] as well as the common sgRNA scaffold primer G00, and the pPLOTv1 blast-mNeonGreen-blast plasmid [51, 86] as DNA template. C9T7 cells were transfected with the amplified templates and tagged cell lines were selected and cloned as described by [50]. The expression of mNG:KINF was confirmed and analysed by fluorescence microscopy and IFC.

### Treatment with flavopiridol

*L. mexicana* C9T7 mNG:KINF promastigotes were grown to mid-log phase before 5 μM flavopiridol (Selleck Chemicals) was added to the medium. Cells were incubated at 27˚C for 12.4 hours before being processed for IFC.

### Immunofluorescence

1 x 10$^7$ *L. mexicana* promastigote mid-log phase cells were washed with phosphate buffered saline (PBS), fixed with 4% paraformaldehyde for 15 mins at room temperature (RT) and

washed again in PBS before being permeabilised in 4% bovine serum albumin (BSA) containing 0.075% Tween-20 for 10 mins at RT. Cells were then sequentially incubated in mouse anti-mNeonGreen (IgG2c; Chromotek), followed by rat anti-mouse IgG(H+L)-AlexaFluor488 (ThermoFisher), mouse anti-KMX-1 (IgG2b; Millipore Merck) and then goat anti-mouse IgG2b(Fcγ)-AlexaFluor594 (ThermoFisher), each for 2 hours at RT. Each antibody was diluted 1:500 in 4% BSA/0.075% Tween-20, and cells were washed twice with PBS between each antibody addition. After the final antibody was washed off, cells were resuspended in PBS and stored at 4˚C in the dark. Before imaging, Fluoroshield™ with DAPI (Sigma) was added to the cell suspension before mounting on poly-L-lysine-coated glass slides. Imaging was performed with a Zeiss Axio Observer 3 microscope and ZEN Blue software (Zeiss). Images were processed with ImageJ version 2.9.0 [87].

## Imaging flow cytometry (IFC)

For IFC of live, unstained cells, ~5 x $10^6$ *L. mexicana* promastigote cells were washed and then resuspended in 60 μl PBS. Alternatively, live cells were first resuspended in 1 ml M199 medium or PBS and stained in the dark with Hoechst 33342 (0.44–1.78 μM for 10 minutes at RT), Vybrant™ DyeCycle™ Orange (DCO; 0.625–10 μM for 30 minutes at RT or 10 minutes at 27˚C), Vybrant™ DyeCycle™ Violet (DCV; 2.5–75 μM for 15 minutes at 27˚C), Vybrant™ Dye-Cycle™ Ruby (DCR; 0.625–25 μM for 15 minutes at 27˚C) or with DRAQ5™ (Invitrogen; 12.5–50 μM for 15 minutes at RT), before being harvested and resuspended in 60 μl PBS. To fix cells, *L. mexicana* were washed in PBS and incubated in either 4% paraformaldehyde (PFA) in PBS for overnight at 4˚C, or in ice-cold 70% methanol/30% PBS overnight at 4˚C. To stain methanol-fixed cells with propidium iodide, cells were washed in PBS/5 mM ethylenediamine-tetraacetic acid (EDTA) and resuspended in 500 μl PBS/5 mM EDTA/10 μgml$^{-1}$ propidium iodide/20 μgml$^{-1}$ RNAse A and incubated at 37˚C for 45 minutes in the dark. All fixed cells were resuspended in 60 μl PBS just prior to being analysed by IFC. Prepared cells were loaded onto an Amnis® ImageStream®X Mk II imaging flow cytometer (Cytek Biosciences) alongside 1 μm diameter SpeedBeads (Amnis) and analysed using INSPIRE™ acquisition software. At least 10,000 events were captured in brightfield (BF) and fluorescence modes at 60X magnification for each sample using laser colour and power settings appropriate for the relevant dye. Cell data were analysed using the post-acquisition IDEAS™ software version 6.0 [88] and with FCS Express™ version 7.10.0007 (DeNovo software by Dotmatics) for cell cycle modelling.

**Gating strategy.** A gating strategy (S12 Fig) was employed to identify single cells and gate out doublets, debris and images with artefacts (e.g. images containing SpeedBeads, out of focus cells, incorrect masking (see below), cell rosettes or more than one cell). Gates were placed accordingly following visual inspection of $\geq 100$ cells corresponding to individual points on the relevant plots for each replicate set. *Leishmania* cells and then singlets were identified from cell body area (calculated *via* the application of a cell body mask (see below) versus side scatter and cell body area versus aspect ratio (AR) plots, respectively. The aspect ratio (minor axis/major axis) of each cell body mask was calculated by IDEAS™ by applying an ellipse of best fit around the mask and determining the longest and shortest dimensions (major and minor axes, respectively). Out of focus images were excluded by gating for cells with a high gradient root mean square (RMS; a measure of pixel contrast), the IDEAS™ Spot Count feature allowed the exclusion of images containing multiple cell masks, and images containing overlapping cells were excluded using cell body area versus compactness plots.

For DCO-labelled and other stained cells, further gating was undertaken to include only cells with high enough fluorescent intensity to have successfully taken up the stain (S13A Fig). For mNG:KINF tagged cell lines, additional gating was performed to identify and mask those

cells expressing mNG:KINF at above background levels. An mNG:KINF negative gate was placed by first analysing untagged cells, before an mNG:KINF positive gate was drawn around mNG:KINF tagged cells with higher fluorescence (S13B Fig). The mNG:KINF fluorescent signal was then masked, as described below, to enable automatic determination of its shape.

**Generation of masks.** Masks were generated for the cell body (excluding the flagellum) to allow calculation of its length, width and area (Fig 1B, 1C), and also for mNG:KINF nuclear staining to allow automatic classification of the staining pattern. The cell body mask was generated from the pre-defined brightfield mask in IDEAS™, using an Adaptive Erosion Coefficient of 71 and an area range of >50 pixels (equivalent to 5.56 μm²). IDEAS™ measures the longest length of the cell body mask at its midline (length) and, using the Thickness Max feature, from the midline to the widest point of the mask (half maximum width), which was then doubled to give the maximum width. As the resolution of measurements was 1 pixel, which equated to 0.33 μm for length and 0.66 μm for width, random noise was added to the data to smooth these measurements. To do this, the data was extracted from IDEAS™ and imported into RStudio (R version 4.0.2), where a random value between -0.33 and 0.33 μm for each length measurement, or between -0.66 and 0.66 μm for each width was added to each data point, before violin plots of the data were generated.

A mask of the mNG:KINF staining was generated using the IDEAS™ Intensity function to identify pixels with a raw intensity of >100. The IDEAS™ spot count feature was used to determine the number of masks per cell, and the aspect ratio intensity (ARI; minor axis intensity/major axis intensity) was used to distinguish between circular masks and spindle-shaped masks, which were determined visually to have ARIs of ≥0.62 and <0.62, respectively.

**Parameters analysed.** The following parameters were automatically measured by IDEAS™: area (of mask; μm²), aspect ratio, brightfield intensity (sum of the intensity of the pixels within a mask, with the background subtracted), circularity (which measures the degree of the mask's deviation from a circle), compactness (similar to circularity but measures all pixels and takes intensity into consideration), height and width (calculated by drawing a rectangular box around the cell and measures its longest and shortest dimensions, respectively), elongatedness (height/width), thickness_min/max (double the distance from the midpoint of the cell to the perimeter at the narrowest/widest point of the cell), length, shape ratio (thickness_min/length), diameter (assumes the object is a circle and calculates the diameter using the area, defined as $2\times\sqrt{(Area/\pi)}$ and the perimeter (of mask, μm).

## Cell cycle analysis

Images of triplicate populations of *Leishmania* C9T7 and C9T7 mNG:KINF promastigote cells stained with DCO were captured by IFC using a laser power of 3 mW, which allowed the detection of DCO but was not sufficient to excite mNG:KINF to detectable levels. Gating was performed as described above and DNA content was examined by plotting a histogram of DCO intensity. Gates were drawn manually around the 2C, intermediate (>2C but <4C) and 4C peaks, with the resultant data averaged for the three replicates. Cells were further classified by their morphology into either long (L; >10 μm) or short (S; ≤10 μm) and wide (W; ≥4 μm) or narrow (N) (< 4 μm) to allow more precise cell cycle stage identification, using previous morphological analysis [3] as a guide. Short cells from the 2C gate were reasoned to be in early G1 phase, while long 2C gated cells were in late G1 phase or early S phase. Long cells in the 4C gate were deemed to be towards the end of S phase or in early G2 phase, while short 4C gated cells were in late G2 phase, mitosis or cytokinesis. Long cells from the intermediate DNA content gate were judged to be mostly in S phase, albeit with some overlap with late G1 and early G2 cells. Additionally, G2 cells (two visible flagella) were distinguished from late S phase cells

(one visible flagellum) by manually determining the number of flagella per cell from IFC brightfield images.

A second aliquot of C9T7 mNG:KINF cells, unstained with DCO, was analysed by IFC using a laser power of 120 mW to permit detection of mNG:KINF. Here, gating was performed as described above and cells were classified according to the fluorescence pattern observed: no fluorescence, one, roughly circular, focus of fluorescence across the nucleus, one elongated focus on the mitotic spindle or two circular foci of fluorescence, one in each of the nuclei present in cells with two nuclei) (S13B Fig). These data were combined with morphological parameters (the length of the cell body and the number of flagella visible in brightfield images) to provide a more precise assessment of the cell cycle stage of individual cells. Short cells with one circular focus of nuclear mNG:KINF fluorescence were deemed to be in very early G1 phase while short cells with no mNG:KINF fluorescence were slightly later (but still early) in G1 phase. Long cells with no mNG:KINF fluorescence were either in late G1 or S phase, while long cells with a circular focus of mNG:KINF in the nucleus were either in S phase or G2 phase, distinguishable by the number of visible flagella per cell. Short wide cells with an elongated, spindle shaped focus of mNG:KINF fluorescence were deemed to be undergoing mitosis, while short wide cells with two nuclei and mNG:KINF fluorescence distributed across both of them had completed mitosis and were undergoing cytokinesis. Three additional unexpected populations were also detected: short wide cells with no mNG:KINF fluorescence (5.9% total cells), long wide cells with two circular mNG:KINF foci (0.9% total cells) and short narrow cells, also with two circular mNG:KINF foci (0.7% total cells). Upon visual inspection of the cells within these populations, the short wide non-fluorescent cells were found to display a wide range of cell body morphologies, many of them aberrant (S14A Fig). The long wide population with two mNG:KINF foci were found to mainly contain two cells located close together that had erroneously been classified as a single cell (S14B Fig), and in the short narrow population with two mNG:KINF foci, one foci was found to be much smaller than usual, suggesting additional mNG:KINF fluorescence in the region of the kinetoplast, independent of their cell cycle stage (S13C Fig). All of these cell populations were discarded from further analysis.

Since cells in late G1 and S phase are of similar length, and the DNA content of late G1 and early S phase cells is practically indistinguishable, in long cells with no mNG:KINF fluorescence, an estimate of the distribution of these cells between late G1 and S phase was made using intensity profiles of DCO fluorescence in DCO stained C9T7 mNG:KINF cells. The percentage of 2C gated cells was calculated, before the percentages of short cells in early S phase (with or without mNG:KINF fluorescence, obtained from IFC of non-DCO stained mNG:KINF cells, as described above) were subtracted, leaving an estimate of cells in late G1 phase. This was then subtracted from the total percentage of long cells with no mNG:KINF fluorescence to provide an estimate of the proportion of S phase cells with no mNG:KINF fluorescence.

Cell cycle timings were calculated using the following equation [3, 53]:

$$x = t\,\frac{\ln(1 - \frac{1}{2}y)}{\ln(2)}$$

where x = time through the cell cycle, t = doubling time of the population and y = cumulative proportion of cells up to and including that stage of the cell cycle. To express cell cycle stage timings as progress through the unit (0–1) cell cycle, *t* was set to 1. This equation accounts for the 'new cell' bias of continual replication within an asynchronous logarithmic population, which results in double the number of newly divided daughter cells being present compared to cells that are about to divide at the end of the previous cell cycle.

## Supporting information

**S1 Fig. Cell cycle-dependent morphologies of *L. mexicana*.** A representation of the morphological changes seen as a cell progresses through the different cell cycle stages. The relative timings of the cell cycle stages are not to scale. M: mitosis; C: cytokinesis; N: nucleus; K: kinetoplast; F: flagellum.
(PDF)

**S2 Fig. Intensity profiles of *L. mexicana* promastigote cells stained with various DNA dyes.** C9T7 cells were incubated with various dyes, as detailed in the Materials and Methods, before being analysed on the ImageStream. Each intensity profile shows the data acquired from ~30,000 cells, with representative images below (left panel: brightfield image; right panel: fluorescence image; scale bars 7 μm). (A) Methanol-fixed cells stained with propidium iodide. (B) Cell cycle modelling of the propidium iodide DNA intensity profile (panel A) using the FCS Express™ Multicycle engine (Rabinovitch & Bagwell debris subtraction [43, 44] and Dean/Jett/Fox cell cycle modelling [45]). The cell cycle stages of each curve are indicated. C-F: live cells stained with three different concentrations of Hoechst, DRAQ5, Vybrant™ DyeCycle™ Violet (DCV) and Vybrant™ DyeCycle™ Ruby (DCR), respectively. (G) Representative images (left panels: brightfield image; right panels: fluorescence image; scale bars 7 μm) of artefacts obtained when staining with suboptimal concentrations of DCO (1.25–10 μM) at room temperature or 27˚C. At a concentration of 10 μM, toxicity was observed with alterations to cell morphology (i) and the appearance of vacuoles (ii). At lower concentrations (1.25–5 μM), suboptimal staining was observed with either just the nucleus (and not the kinetoplast) staining (iii) or staining being observed additionally in the cytoplasm and/or cell membrane (iv).
(PDF)

**S3 Fig. DCO does not affect viability of *L. mexicana* promastigotes.** Following incubation with DCO, C9T7 cells were washed three times in PBS and resuspended in M199 at a density of 5 x 10$^5$ cellsml$^{-1}$. Control (unstained) cells were similarly processed. Cell density was measured over 72 hours and a growth curve plotted.
(PDF)

**S4 Fig. Stability of DCO.** (A) Triplicate populations of *L. mexicana* promastigote C9T7 cells were stained with 0.625 μM DCO for 30 minutes at room temperature before being resuspended in PBS. Following resuspension in PBS, cell samples were analysed by IFC at the time points indicated. Fluorescence intensity profiles are shown for one representative replicate (*n* = ~15,000 cells). (B) Cell cycle modelling of the fluorescence intensity profiles presented in (A) using FCS Express™.
(PDF)

**S5 Fig. Morphological parameters of different cell cycle stages.** *L. mexicana* C9T7 promastigotes were stained with DCO, analysed by IFC and gated according to their DNA intensity (2C, intermediate DNA content (Int.; 2C-4C) and 4C) (Fig 3A). The area, aspect ratio (AR), brightfield intensity (BF_Intensity), circularity, compactness, diameter, elongatedness, height, perimeter, shape ratio and thickness_min of the cells were then plotted for each gate, as indicated in the figure.
(PDF)

**S6 Fig. Linking DNA content and cellular dimensions.** (A) Length and (B) width distributions of C9T7 cells from the DNA intensity plot presented in Fig 3A. Top panels: cells within the 2C gate; middle panels: cells within the intermediate (2C-4C) gate; bottom:

overlay of cells in all gates.
(PDF)

**S7 Fig. Morphological parameters of cells gated on DNA content and length.** *L. mexicana* C9T7 promastigotes were stained with DCO, analysed by IFC and gated according to their DNA intensity (2C, intermediate DNA content (Int.; 2C-4C) and 4C) (Fig 3A). 4C gated cells were then further gated according to their length (short: <10 μm; long ≥ 10 μm) (Fig 3B). The area, aspect ratio (AR), brightfield intensity (BF_Intensity), circularity, compactness, diameter, elongatedness, height, length, perimeter, shape ratio, thickness_min and width of the cells in each gate were then plotted, as indicated in the figure.
(PDF)

**S8 Fig. mNG:KINF displays a cell cycle-dependent localisation.** Immunofluorescence was performed on *L. mexicana* promastigote C9T7 (control) (A) or C9T7 mNG:KINF cells (B-I) with the KMX-1 antibody to detect β-tubulin and an anti-mNG antibody to detect mNG: KINF. Cells were also stained with DAPI to visualise DNA. From left to right: brightfield, DAPI (blue), KMX-1 (red), mNG (green) and DAPI/KMX-1/mNG merged images. The number of nuclei (N), kinetoplasts (K) and flagella (F) per cell is indicated (N^d indicates a dividing nucleus), along with the cell cycle stage and mNG:KINF fluorescence pattern. White arrowheads indicate N, K and F. Dotted boxes highlight the mitotic spindle as stained by the KMX-1 antibody, which is enlarged and indicated by asterisks in the solid white line boxes; green arrowheads indicate mNG:KINF staining within the nucleus. Scale bar = 5 μm.
(PDF)

**S9 Fig. Dimensions of G2 phase cells.** The lengths (A) and widths (B) of C9T7 mNG:KINF cells identified as being in G2 phase were plotted.
(PDF)

**S10 Fig. S phase cells with elongated mNG:KINF fluorescence.** Some long, narrow C9T7 mNG:KINF cells with a single flagellum were observed to possess a single elongated focus of mNG:KINF fluorescence that led to them being classified automatically as having a mitotic spindle. Cells such as these were manually removed from the 'mNG:KINF spindle' population and added to the S phase 1CF population.
(PDF)

**S11 Fig. Excitation and measurement of DCO fluorescence in C9T7 mNG:KINF cells.** C9T7 mNG:KINF cells were stained with DCO before being imaged with IFC using a 488 nm laser with a power of 3 mW to excite only DCO and not mNG:KINF to detectable levels. Fluorescence emitting in channel 3 (560–595 nm) was measured. (A) Raw max pixel intensity plots (the fluorescence intensity of the brightest pixel in each image) of (i) unstained C9T7, (ii) unstained C9T7 mNG:KINF and (iii) DCO-stained C9T7 mNG:KINF cells. (B) Example DCO fluorescence intensity plot of C9T7 mNG:KINF cells stained with DCO. 2C, intermediate (2C-4C) and 4C gates were placed as indicated (blue boxes) with the % cells within each of the gates indicated on the right. Data was from the Run 2 replicate; data from the Runs 1 and 3 replicates is in S1G Table.
(PDF)

**S12 Fig. ImageStream cell gating strategy.** Following ImageStream acquisition, data were processed using a user-defined gating strategy in IDEAS to identify single cells and gate out doublets, debris and images with artefacts. First, cell body area was plotted against side scatter, with gating applied to exclude debris and speed beads (A; teal gate). Next, singlet cells were selected using a cell body area vs aspect ratio plot (B; blue gate). The best in focus cells in bright

field (BIF_BF) were identified via the Gradient_RMS parameter (C; orange bar). Images with a single mask (i.e. one object per image) were selected using "Spot Count" (D) and plotting cell body area against cell compactness enabled the removal of overlapping cells, resulting in the 'Processed' population (E, light blue gate), which was then utilised for morphological and fluorescence analysis. $n$ = 30,000 C9T7 cells.
(PDF)

**S13 Fig. ImageStream cell cycle gating strategy.** Using a 'processed' population (S11 Fig), separate gating strategies were employed for the cell cycle analysis of DCO-stained C9T7 mNG:KINF cells excited with a laser power of 3 mW (A) and unstained C9T7 mNG:KINF cells excited with a laser power of 120 mW (B). (A) The best in-focus DNA images were selected using Gradient_RMS (i; red bar) and cells sufficiently stained with DCO were identified via their fluorescent intensity (ii; orange gate). The cell cycle stages of the cells were then determined based on their DCO intensity (iii; G1, S and G2/M bars). An equivalent gating strategy was employed for cells stained with other dyes (Hoechst, DRAQ5, DCV and DCR). To facilitate further resolution of cell cycle stage within the G2/M cell population of DCO-stained cells, morphological characteristics were analysed. Cell body length (iv) and width (v; using the Thickness Max parameter) were plotted to identify short (length ≤10 μm), long (length >10 μm), narrow (width <4 μm) and wide (width ≥4 μm) cells. (B) Cells expressing mNG:KINF were identified by gating on green fluorescence greater than that of the parental untagged cell line (i; green gate). Cell cycle stages were determined from the cell length (ii, short and long bars, as in (A)) and width (iii; Thickness Max, narrow and wide bars, as in (A)) and by their mNG:KINF fluorescence pattern. mNG:KINF foci were identified using masks, allowing the separation of cells with one or two mNG:KINF foci (iv). The aspect ratio intensity of cells with a single mNG:KINF focus allowed cells with a circular focus or an elongated spindle-shaped focus to be distinguished (v).
(PDF)

**S14 Fig. Aberrant mNG:KINF populations.** Example images of C9T7 mNG:KINF cells that were discarded from further analysis. (A) short, wide cells with mNG fluorescence below the threshold. The majority of these cells had aberrant morphology that meant they could not be assigned to any given cell cycle stage. (B) cells assigned by IDEAS[TM] as long and wide with two circular mNG:KINF foci. The majority of images from this group were found to be of two individual cells in close proximity that had been misclassified. (C) short narrow cells that were classified as having two circular mNG:KINF foci by IDEAS[TM]. All of these cells were found to have unequally sized foci, suggesting that there was some additional fluorescence in the region of the kinetoplast (white arrowheads).
(PDF)

**S1 Table. Raw and compiled IFC data.** (A) Live vs fixed raw data. Length and width measurements used to compile Fig 1D. (B) DCO vs PI percentages. Cell cycle modelling data from triplicate IFC analysis of C9T7 cells stained with DCO or PI as used to compile Fig 2D. (C-E) KINF Run1-3. Raw and processed data from triplicate IFC analysis of unstained mNG:KINF cells analysed with a laser power of 120 mW. Further details are given in the 'Notes' section at the bottom of the worksheet. (F) KINF averages. Mean values + standard deviations for data from the three mNG:KINF replicates, used to compile Fig 5B and 5D. (G) DCO. Gate data from triplicate IFC analysis of DCO-stained mNG:KINF cells analysed with a laser power of 3 mW. Run 2 data is presented in S10 Fig. (H) Calculated timings. Compiled mNG:KINF and DCO IFC data used to generate the cell cycle stage durations presented in Table 1 and Fig 7. Further details are given in the 'Notes' section at the bottom of the worksheet. (I) Cytokinesis

furrowing. Raw data and average values from visual classification of furrows in mNG:KINF cells with spindles or 2CF (3 replicates) used to compile Fig 6A. (J) Flavopiridol. Raw and processed flow cytometry and IFC data from duplicate C9T7 mNG:KINF cultures treated or not with 5 μgml⁻¹ flavopiridol for 12.4 hrs used to compile Fig 7A and 7B. For flow cytometry analysis, cells were fixed in MeOH and stained with PI, and data collected was gated according to DNA content (2C, Int. 4C). For IFC, the mNG:KINF staining pattern was classified using the processing pipeline and gating strategy described above.
(XLSX)

## Acknowledgments

The authors would like to thank Pj Chana from Cytek Bio for useful advice, discussions and training on IFC.

## Author Contributions

**Conceptualization:** Jessie Howell, Sulochana Omwenga, Melanie Jimenez, Tansy C. Hammarton.

**Data curation:** Jessie Howell, Sulochana Omwenga, Tansy C. Hammarton.

**Formal analysis:** Jessie Howell, Sulochana Omwenga, Melanie Jimenez.

**Funding acquisition:** Melanie Jimenez, Tansy C. Hammarton.

**Investigation:** Jessie Howell, Sulochana Omwenga, Tansy C. Hammarton.

**Methodology:** Jessie Howell, Sulochana Omwenga, Melanie Jimenez, Tansy C. Hammarton.

**Project administration:** Melanie Jimenez, Tansy C. Hammarton.

**Resources:** Melanie Jimenez, Tansy C. Hammarton.

**Software:** Jessie Howell, Melanie Jimenez.

**Supervision:** Melanie Jimenez, Tansy C. Hammarton.

**Validation:** Jessie Howell, Sulochana Omwenga, Tansy C. Hammarton.

**Visualization:** Jessie Howell, Sulochana Omwenga, Tansy C. Hammarton.

**Writing – original draft:** Jessie Howell, Tansy C. Hammarton.

**Writing – review & editing:** Jessie Howell, Sulochana Omwenga, Melanie Jimenez, Tansy C. Hammarton.

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
