## [Decision Letter · Decision Letter 0]

18 Jun 2024

PONE-D-24-20727Analysis of the *Leishmania mexicana* promastigote cell cycle using imaging flow cytometry provides new insights into cell cycle flexibility and events of short duration.PLOS ONE

Dear Dr. Hammarton,

Thank you for submitting your manuscript to PLOS ONE. After careful consideration, we feel that it has merit but does not fully meet PLOS ONE’s publication criteria as it currently stands. Therefore, we invite you to submit a revised version of the manuscript that addresses the points raised during the review process.

**This manuscript presents a state-of-the art, IFC-based pipeline to combine the population-level and quantitative analysis advantages of flow cytometry to bring to bear on analyses of the Leishmania cell cycle in live cells. One advantage of this high-throughput approach, in principle, is to allow capture of hard-to-identify populations of cells transiting through short-lived sectors of the cell-cycle. The authors claim several novel observations: including a newly detailed description of the Leishmania promastigote cell cycle and establishing the durations of distinct cell cycle phases. The authors also present a potentially useful methodological proof of principle for IFC in evaluation of the action of anti-Leishmania drug candidates, as they may relate to the cell cycle. There appear, however, to be a several caveats in their approach and findings (see reviewer comments), that need to be addressed.**

**The authors must please respond point-by-point to each of the reviewers’ comments. For clarification, they do need to respond to Reviewer 2’s concerns regarding the anti-tubulin staining.**

**The authors used the terms “elongatedness” and “length” (e.g. line 220); it is unclear whether these terms are referring to the same or different parameters. For reviewer 2, please also clarify whether the cell cycle analyses in Figs 2 and 3 were performed on mNG-KINF-expressing cells, or not .**

**Some minor issues with language and typos were also noted:**

**e.g.s**

**Line 17: “Promastigote Leishmania mexicana have…”**

**Line 371: “Visually inspected for the presence of two visible flagella” (remove ”Visually” or “visible”).**

We look forward to receiving your revised manuscript.

Kind regards,

Ben L. Kelly, Ph.D.

Academic Editor

PLOS ONE

Journal Requirements:

"This work was supported by the Engineering and Physical Sciences Research Council (PhD studentship no. EP/R513222/1 held by J.H.), the Royal Academy of Engineering (Research Fellowship no. RF/201718/1741 awarded to M.J.). and the Cunningham Trust (PhD studentship grant (PhD-CT-19-14) awarded to T.H. and M.J., and held by S.O.)."

"The authors would like to thank Pj Chana from Cytek Bio for useful advice, discussions and training on IFC. This work was supported by the Engineering and Physical Sciences Research Council (PhD studentship no. EP/R513222/1 held by J.H.), the Royal Academy of Engineering (Research Fellowship no. RF/201718/1741 awarded to M.J.). and the Cunningham Trust (PhD studentship grant (PhD-CT-19-14) awarded to T.H. and M.J., and held by S.O.)."

"This work was supported by the Engineering and Physical Sciences Research Council (PhD studentship no. EP/R513222/1 held by J.H.), the Royal Academy of Engineering (Research Fellowship no. RF/201718/1741 awarded to M.J.). and the Cunningham Trust (PhD studentship grant (PhD-CT-19-14) awarded to T.H. and M.J., and held by S.O.)."

Reviewers' comments:

Reviewer's Responses to Questions

**Comments to the Author**

1. Is the manuscript technically sound, and do the data support the conclusions?

Reviewer #1: Yes

Reviewer #2: Partly

2. Has the statistical analysis been performed appropriately and rigorously? 

Reviewer #1: Yes

Reviewer #2: Yes

3. Have the authors made all data underlying the findings in their manuscript fully available?

Reviewer #1: Yes

Reviewer #2: No

4. Is the manuscript presented in an intelligible fashion and written in standard English?

Reviewer #1: Yes

Reviewer #2: Yes

5. Review Comments to the Author

Reviewer #1: The paper by Howell et al details a new method for analyzing the cell cycle in L. Mexicana using imaging flow cytometry. The paper is an important contribution because analyzing cell cycle using conventional microscopy methods is very labor intensive and looking for rare phenotypes can be challenging with this technique. While conventional DNA staining can separate cells into G1, S, and G2, the addition of images combined with the flow cytometry technique allows further delineation for stages of the cell cycle based on length and width measurements along with other morphological characteristics so that early and late G1 phase cells, S phase, and G2/M/cytokinesis cells can be distinguished. In addition, using a neon green KINF reporter cell line can help to distinguish cells in the mitotic stage from G2 cells. Imaging flow cytometry allows for much higher throughput and less labor intensive cell cycle analysis. These methods will be especially useful when analyzing the effects of various drugs or other genetic manipulations within the L. Mexicana system. Using a combination of IFC techniques, the authors were able to come up with detailed time estimates for how L. mexicana progresses through the different stages of the cell cycle. The paper is well written and clear, and most of my comments are minor suggestions. While I plan to enumerate a few suggestions in the ‘major’ category, the authors should not feel bound to follow them; in my opinion the paper is publishable pretty much as is.

Major points

• I’m not entirely sure that the data presented in Figure 4 is necessary. There is quite a bit of background staining here for the neon green, as shown in the control cells that do not have the construct, while the IFC imaging looks comparatively much cleaner. While the authors use a beta-tubulin stain for these conventional microscopy images, this stain doesn’t feature much at all within the text and doesn’t seem necessary for making any of their main points. The spindle staining shown in Figure 4H and the double stain shown in Figure 4I are clearly visible in Figure 5A. The lack of KINF staining in G1/S cells is also more clearly seen in the IFC analysis because there is less background. For this reason, Figure 4 doesn’t seem to contribute much to the big picture.

• Figure 8 presents some nice data on a drug previously shown to alter the cell cycle. The authors show an arrest at the 4C stage using conventional PI staining, and then further analyze the cells using the KINF neon green marker to show that the cells get stuck before mitosis (if I’m interpreting this right). I found myself wondering how the conventional PI stain for drug treated cells compares to the dicycle orange stain that the authors so nicely showed earlier. I realize that the dicycle orange stain is not compatible with the neon green marker, so it might be necessary to get additional data to show how these two analyses compare. While I don’t think this is absolutely necessary, it might be a nice addition to the paper if it is not too labor intensive. It would help to show that the dicycle orange analysis can still be useful for analyzing drug-related cell cycle defects in Leishmania lines that do not carry the KINF marker, and that it’s comparable to the PI analysis that everyone is very familiar with in cases where the cell cycle is perturbed.

• The authors present two main techniques for analyzing the cell cycle with IFC, one with dyecycle orange and one with the KINF marker. If I understand the data correctly, these techniques can’t be performed at the same time because of spectral overlap. Given that this is the case, it might be nice for the authors to help guide readers as to which technique is the most useful for a particular experimental study. For example, those interested in characterizing defects in G2/M might find the KINF staining more useful, whereas people interested in distinguishing problems entering S phase might be able to rely on the length measurements combined with the dicycle orange stain.

Minor points

• In Figure 2 the figure legend makes the point that the nucleus divides first, that point should be made in the main text

• Figure 3B, it would be helpful to put in a title that marks these cells specifically in the 4C gate

• Figure 4, label the stain as beta-tubulin above KMX1 antibody label

• Should 4D also say mNG-ve? Stay consistent with these labels

• In Figure 4, it would be nice if G, and H also had green arrowheads to indicate positive staining to be consistent

• Figure 6 it would be helpful to put a legend in a corner indicating what arrows and stars mean so that readers don’t have to go hunting in the legend

• It would be helpful to summarize the specific contribution of the neon green at the end of that section, it lets us distinguish between this and that

• Figure 8 please put full name of drug in figure

Reviewer #2: This manuscript describes a new cell cycle analysis pipeline for live Leishmania mexicana that utilizes state of the art flow cytometry that is directly coupled to both phase contrast microscopy and fluorescence intensity microscopy. This allows very rapid analysis of large numbers of live cells on the basis of three different imaging modalities thus providing a large saving in time and labor. The authors argue that this type of analysis can reduce artifacts while greatly increasing the efficiency of experiments designed to analyze mutant or drug-induced perturbations in the L. mexicana cell cycle.

While the data and procedures that are the primary focus of this manuscript are more suitable to a protocol manuscript, this submission is as a field research report. Unlike a typical field research report, this manuscript does not present a testable biological hypothesis or propose to test an important new finding using new methodology, but rather purports to discover and list technical advantages of a cutting edge flow cytometry instrument.

In this sense, this report is one-sided in primarily only providing evidence in support of the protocols used in this approach and does not discuss or provide balanced insight to the apparent weaknesses and compromises inherent in the protocols.

Furthermore, this report documents details of protocol outcomes that are purported to be important first-of-their kind conclusions.

1) This report is not the first imaging flow cytometry analysis of Leishmania. Nor the first to make correlations of cell cycle state to morphology albeit with previously published reports that did not utilize IFC.

2) Necessity of using very high laser power (120mW vs 3mW for DNA stains) to detect mNeon GFP and inability to detect mCherry fusions at all. This is a major limitation of this instrument which should be discussed and experimentally assessed as many investigators rely on GFP/RFP-based fluorescence in live cell analyses. It seems that IFT can detect broad or very bright Leishmania fluorescence, it is not capable or barely capable of detecting organellar fluorescence. Had the authors succeeded in finding a two-channel solution, perhaps mScarlet3 fusions, to use with mNeon, this report would have been more compelling.

3) Lack of dose response data to justify the stated DNA probe concentrations prior to IFC. Raw DCO data are not included in Figure S2. DCO data presented in Figures 2 and 3 appear to have undergone further gating and filtering to enrich for high-staining cells; a procedure not utilized for the other DNA stains. The conclusion that the other stains are not quantitative is not based on thorough experimental evidence.

4) Cell cycle analysis using DCO (Figures 2 and 3; DCO+ in Fig S12) were performed on mNG transfected cells. It is argued that although DCO and mNG share overlapping excitation/emission profiles, that mNG did not contribute to these data. Given that there is stated variability in mNG expression levels and variability in DCO staining strength (which required gating and filtering), it remains distinctly possible that mNG materially contributed to these spectra (especially the lower intensity data). This experiment should be performed on cells without mNG.

5) DCO raw intensity curves for data in the supplementary excel table are not provided. Without the curves, the relative qualities of the peak intensity values recorded in the tables cannot readily be assessed. The curves should be in the SI.

6) The discovery of pleiotropic cleavage furrow progression is weakened by lack of live cell time-lapse corroboration with the IFC data, plus worrying artifacts of partially -of-focus cleavage furrows that populate the dataset. The limited depth of field of the microscopy reduces the confidence of the direction of furrow formation conclusions, especially since they disagree with prior published results.

7) The DCO fluorescence intensity decay spectra in Figure S4 do not support the conclusion that there is significant decay in the peak signal over 46 minutes; peak signal remains unchanged by visual inspection (2.5 x 10fourth) but there is an obvious rightward shift of reduced intensity for the smaller second peak. This calls into question whether the second smaller peak represents nuclear DNA while the major peak represents kinetoplast DNA.

8) KMX1 tubulin staining appears very poor in Figure 4, with the mitotic spindle barely resolved from background. It could be argued that the signal to noise of the mNG KINF signal allows better visualization of the morphogenesis of the mitotic spindle. Furthermore, KINF staining within the nuclear compartment in interphase prevents a clear assessment of the G2/M transition point as the initiation of spindle morphogenesis cannot be marked; perhaps centrin or similar could be used to mark G2/M with the advent of centrosome duplication (but it is unlikely that IFC could detect centrosome staining).

9) Gating strategy (S11) relied on ‘visual inspection’ of ~100 cells corresponding to individual points… Please confirm in the methods that each replicate set relied on independently assessed visual gating.

10) While the use of flow cytometry to evaluate the cell cycle stage blocked by flavopiridol (G2/M) has been previously published, new here is a correlation to mNG KINF localization and cell morphology. While it can be argued that flavopiridol blocks cell cycle progression at G2/M if mNG marks nascent spindle microtubule formation, or rather G2 alone as the authors argue, there is little new information here. Given the promiscuous inhibitory behavior of flavopiridol, this result is unlikely to be of broad interest.

6. PLOS authors have the option to publish the peer review history of their article (what does this mean?). If published, this will include your full peer review and any attached files.

Reviewer #1: **Yes: **Danae Schulz

Reviewer #2: No

---

## [Author Response · Author response to Decision Letter 0]

30 Aug 2024

Response to reviewers’ comments – Howell et al Manuscript number: PONE-D-24-20727

We would like to thank the two reviewers and editor for their thorough and helpful reviews of our manuscript. We have detailed our responses to their comments below.

>This manuscript presents a state-of-the art, IFC-based pipeline to combine the population-level and quantitative analysis advantages of flow cytometry to bring to bear on analyses of the Leishmania cell cycle in live cells. One advantage of this high-throughput approach, in principle, is to allow capture of hard-to-identify populations of cells transiting through short-lived sectors of the cell-cycle. The authors claim several novel observations: including a newly detailed description of the Leishmania promastigote cell cycle and establishing the durations of distinct cell cycle phases. The authors also present a potentially useful methodological proof of principle for IFC in evaluation of the action of anti-Leishmania drug candidates, as they may relate to the cell cycle. There appear, however, to be a several caveats in their approach and findings (see reviewer comments), that need to be addressed.

The authors must please respond point-by-point to each of the reviewers’ comments. For clarification, they do need to respond to Reviewer 2’s concerns regarding the anti-tubulin staining.

>The authors used the terms “elongatedness” and “length” (e.g. line 220); it is unclear whether these terms are referring to the same or different parameters. 

These are distinct parameters. Elongatedness is height/width and is defined in the methods (lines 885-886). However, to aid clarity, we have added this definition at first use to the main text (line 233).

>For reviewer 2, please also clarify whether the cell cycle analyses in Figs 2 and 3 were performed on mNG-KINF-expressing cells, or not .

 As already stated in the legends to these figures, these data were from analyses performed with C9T7 cells (ie parental not mNG-KINF). We have added ‘C9T7’ to the description in the Fig 2B legend (line 200) to clarify that these analyses were of the C9T7 cells presented in Fig 2A images. We have also added ‘C9T7’ references in the main text description (lines 149 and 226) to avoid confusion.

Some minor issues with language and typos were also noted:

e.g.s

Line 17: “Promastigote Leishmania mexicana have…”

This is intended as a plural reference to all Leishmania mexicana promastigote parasites and is therefore correct as it stands.

Line 371: “Visually inspected for the presence of two visible flagella” (remove ”Visually” or “visible”).

We have swapped ‘visually inspected’ for ‘inspected by eye’. It is important to highlight that this analysis was not done by computer, and to keep the ‘visible’ flagella description – this refers to flagella that have emerged from the flagellar pocket and can be seen by light microscopy – some cells will have internal flagella that have started to form but haven’t yet emerged from the cell and will therefore not be visible by eye (now line 377).

We look forward to receiving your revised manuscript.

Kind regards,

Ben L. Kelly, Ph.D.

Academic Editor

PLOS ONE

Journal Requirements:

"This work was supported by the Engineering and Physical Sciences Research Council (PhD studentship no. EP/R513222/1 held by J.H.), the Royal Academy of Engineering (Research Fellowship no. RF/201718/1741 awarded to M.J.). and the Cunningham Trust (PhD studentship grant (PhD-CT-19-14) awarded to T.H. and M.J., and held by S.O.)."

Done.

"The authors would like to thank Pj Chana from Cytek Bio for useful advice, discussions and training on IFC. This work was supported by the Engineering and Physical Sciences Research Council (PhD studentship no. EP/R513222/1 held by J.H.), the Royal Academy of Engineering (Research Fellowship no. RF/201718/1741 awarded to M.J.). and the Cunningham Trust (PhD studentship grant (PhD-CT-19-14) awarded to T.H. and M.J., and held by S.O.)."

"This work was supported by the Engineering and Physical Sciences Research Council (PhD studentship no. EP/R513222/1 held by J.H.), the Royal Academy of Engineering (Research Fellowship no. RF/201718/1741 awarded to M.J.). and the Cunningham Trust (PhD studentship grant (PhD-CT-19-14) awarded to T.H. and M.J., and held by S.O.)."

Done.

Reviewers' comments:

Reviewer's Responses to Questions

Comments to the Author

1. Is the manuscript technically sound, and do the data support the conclusions?

Reviewer #1: Yes

Reviewer #2: Partly

2. Has the statistical analysis been performed appropriately and rigorously? 

Reviewer #1: Yes

Reviewer #2: Yes

3. Have the authors made all data underlying the findings in their manuscript fully available?

Reviewer #1: Yes

Reviewer #2: No

4. Is the manuscript presented in an intelligible fashion and written in standard English?

Reviewer #1: Yes

Reviewer #2: Yes

5. Review Comments to the Author

Reviewer #1: The paper by Howell et al details a new method for analyzing the cell cycle in L. Mexicana using imaging flow cytometry. The paper is an important contribution because analyzing cell cycle using conventional microscopy methods is very labor intensive and looking for rare phenotypes can be challenging with this technique. While conventional DNA staining can separate cells into G1, S, and G2, the addition of images combined with the flow cytometry technique allows further delineation for stages of the cell cycle based on length and width measurements along with other morphological characteristics so that early and late G1 phase cells, S phase, and G2/M/cytokinesis cells can be distinguished. In addition, using a neon green KINF reporter cell line can help to distinguish cells in the mitotic stage from G2 cells. Imaging flow cytometry allows for much higher throughput and less labor intensive cell cycle analysis. These methods will be especially useful when analyzing the effects of various drugs or other genetic manipulations within the L. Mexicana system. Using a combination of IFC techniques, the authors were able to come up with detailed time estimates for how L. mexicana progresses through the different stages of the cell cycle. The paper is well written and clear, and most of my comments are minor suggestions. While I plan to enumerate a few suggestions in the ‘major’ category, the authors should not feel bound to follow them; in my opinion the paper is publishable pretty much as is.

We thank the reviewer for their positive review of our manuscript. 

Major points

• I’m not entirely sure that the data presented in Figure 4 is necessary. There is quite a bit of background staining here for the neon green, as shown in the control cells that do not have the construct, while the IFC imaging looks comparatively much cleaner. While the authors use a beta-tubulin stain for these conventional microscopy images, this stain doesn’t feature much at all within the text and doesn’t seem necessary for making any of their main points. The spindle staining shown in Figure 4H and the double stain shown in Figure 4I are clearly visible in Figure 5A. The lack of KINF staining in G1/S cells is also more clearly seen in the IFC analysis because there is less background. For this reason, Figure 4 doesn’t seem to contribute much to the big picture.

We agree that the β-tubulin immunofluorescent staining in the fixed cells is not as high quality as the live cell fluorescence imaging by IFC, but we believe it is important to include as a control to demonstrate that the mNG:KINF signal colocalises with the mitotic spindle, as expected. Unfortunately, Fig 5A doesn’t show this colocalisation – the two panels in this figure show the mNG:KINF signal and the automatic mask applied to it, rather than a β-tubulin signal, so don’t provide this necessary control. We tried hard to optimise the conditions for the anti-mNG staining but without success. However, we only use this to show that there is mNG:KINF signal at the spindle, and the rest of our analysis of mNG:KINF localisation was performed with live cells and IFC, so do not believe this detracts from our interpretation of mNG:KINF cell cycle distribution. In view of the reviewers’ comments about this figure, we have moved it to the Supplementary data (S8 Fig) so that it doesn’t detract from the main story of the manuscript.

• Figure 8 presents some nice data on a drug previously shown to alter the cell cycle. The authors show an arrest at the 4C stage using conventional PI staining, and then further analyze the cells using the KINF neon green marker to show that the cells get stuck before mitosis (if I’m interpreting this right). I found myself wondering how the conventional PI stain for drug treated cells compares to the dicycle orange stain that the authors so nicely showed earlier. I realize that the dicycle orange stain is not compatible with the neon green marker, so it might be necessary to get additional data to show how these two analyses compare. While I don’t think this is absolutely necessary, it might be a nice addition to the paper if it is not too labor intensive. It would help to show that the dicycle orange analysis can still be useful for analyzing drug-related cell cycle defects in Leishmania lines that do not carry the KINF marker, and that it’s comparable to the PI analysis that everyone is very familiar with in cases where the cell cycle is perturbed.

Given that DCO quantitation of DNA in live cells is equivalent to PI quantitation of DNA in fixed cells (Fig 2B-D and S2A,B Fig), we see no reason that DCO staining on its own could not be used for investigating cell cycle defects following drug treatment. Our prediction is that this would allow changes to the 2C, intermediate and 4C DNA fluorescence peaks, or the appearance of abnormal peaks (eg 8C for polyploid cells) to be identified, in the same manner as PI does in fixed cells, although testing this was not possible within the time-constraints of this project (the first two authors are now writing up their PhDs). Additionally, the ability to combine DCO staining of live cells with analysis of cell morphology should allow greater resolution of some cell cycle stages over PI (e.g. compare DCO staining alone (equivalent to PI) and DCO staining + cell morphology in new summary Table 2 – see below). However, like PI, without employing KINF fluorescence, DCO staining alone would not provide any information on mitotic spindle formation, and even if combined with morphological analysis, could not distinguish G2 cells from M phase cells. We therefore hope that in future a different fluorescent marker compatible with DCO could be employed to tag KINF to enable simultaneous detection.

• The authors present two main techniques for analyzing the cell cycle with IFC, one with dyecycle orange and one with the KINF marker. If I understand the data correctly, these techniques can’t be performed at the same time because of spect

---

## [Decision Letter · Decision Letter 1]

19 Sep 2024

Analysis of the *Leishmania mexicana* promastigote cell cycle using imaging flow cytometry provides new insights into cell cycle flexibility and events of short duration.

PONE-D-24-20727R1

Dear Dr. Hammarton,

We’re pleased to inform you that your manuscript has been judged scientifically suitable for publication and will be formally accepted for publication once it meets all outstanding technical requirements.

Kind regards,

Ben L. Kelly, Ph.D.

Academic Editor

PLOS ONE

Additional Editor Comments (optional):

Reviewers' comments:

Reviewer's Responses to Questions

**Comments to the Author**

1. If the authors have adequately addressed your comments raised in a previous round of review and you feel that this manuscript is now acceptable for publication, you may indicate that here to bypass the “Comments to the Author” section, enter your conflict of interest statement in the “Confidential to Editor” section, and submit your "Accept" recommendation.

Reviewer #1: All comments have been addressed

Reviewer #2: All comments have been addressed

2. Is the manuscript technically sound, and do the data support the conclusions?

Reviewer #1: Yes

Reviewer #2: Yes

3. Has the statistical analysis been performed appropriately and rigorously? 

Reviewer #1: Yes

Reviewer #2: Yes

4. Have the authors made all data underlying the findings in their manuscript fully available?

Reviewer #1: Yes

Reviewer #2: Yes

5. Is the manuscript presented in an intelligible fashion and written in standard English?

Reviewer #1: Yes

Reviewer #2: Yes

6. Review Comments to the Author

Reviewer #1: The authors have addressed all my concerns, thank you! I'm happy for the manuscript to be published as is.

Reviewer #2: The authors have addressed all the major concerns cited in my initial review and have made clear the limitations of their study. The high throughput method outlined in detail by the authors will open the door to new approaches to study the biology of the cell cycle in this organism, such as screening for much-needed mitotic inhibitors and as an aid towards elucidating their mechanism(s). While I fully support the publication of this report as an improved method with greater resolution to study the cell cycle biology of this organism with greater precision, I hesitate to classify this work as a hypothesis-driven research report. It reads more as a set of validation tests in preparation for an upcoming small molecule screen; possibly more appropriate as a protocol rather than a research report.

7. PLOS authors have the option to publish the peer review history of their article (what does this mean?). If published, this will include your full peer review and any attached files.

Reviewer #1: **Yes: **Danae Schulz

Reviewer #2: No

---

## [Editor Report · Acceptance letter]

24 Sep 2024

PONE-D-24-20727R1 

PLOS ONE

Dear Dr. Hammarton, 

I'm pleased to inform you that your manuscript has been deemed suitable for publication in PLOS ONE. Congratulations! Your manuscript is now being handed over to our production team.

Kind regards, 

on behalf of

Dr. Ben L. Kelly 

Academic Editor

PLOS ONE